



# 1 Drivers of cloud droplet number variability in the summertime
# 2 Southeast United States

Aikaterini Bougiatioti[1,2], Athanasios Nenes[2,3,4], Jack J. Lin[2,a], Charles A. Brock[5], Joost A. de Gouw[5,6,b], Jin
Liao[5,6,c,d], Ann M. Middlebrook[5], André Welti[5,6,e]
[1]Institute for Environmental Research & Sustainable Development, National Observatory of Athens, P.
Penteli, GR-15236, Greece
[2]School of Earth & Atmospheric Sciences, Georgia Institute of Technology, Atlanta, GA 30332, USA
[3]Laboratory of Atmospheric Processes and their Impacts, School of Architecture, Civil & Environmental
Engineering, École Polytechnique Fédérale de Lausanne, CH-1015, Lausanne, Switzerland
[4]Institute for Chemical Engineering Sciences, Foundation for Research and Technology Hellas, Patras, GR-
26504, Greece
[5]Chemical Sciences Division, NOAA Earth System Research Laboratory, Boulder, CO, 80305, USA
[6]Cooperative Institute for Research in Environmental Sciences, Univ. of Colorado, Boulder, CO, 80309,
USA
[a] now at: Nano and Molecular Systems Research Unit, Box 3000, FI-90014 University of Oulu, Oulu,
Finland
[b] now at: Department of Chemistry and Biochemistry, University of Colorado Boulder, Boulder, CO, USA
[c] now at: Atmospheric Chemistry and Dynamic Laboratory, NASA Goddard Space Flight Center,
Greenbelt, MD, USA
[d] now at: Universities Space Research Association, GESTAR, Columbia, MD, USA
[e] now at: Atmospheric Composition Research Unit, Finnish Meteorological Institute, Helsinki, Finland
*Correspondence to*: Aikaterini Bougiatioti (abougiat@noa.gr), Athanasios Nenes
(athanasios.nenes@epfl.ch).

## 24 Abstract

The Southeast United States has experienced a different climate warming trend compared to other places
worldwide. Several hypotheses have been proposed to explain this trend, one being the interaction of
anthropogenic and biogenic aerosol precursors that synergistically promote aerosol formation, elevate cloud
droplet concentration and induce regional cooling. We examine these aerosol-cloud droplet links by
analyzing regional scale data collected onboard the NOAA WP-3D aircraft during the 2013 Southeast
Nexus (SENEX) campaign to quantify the sensitivity of droplet number to aerosol number, chemical
composition and vertical velocity on a regional scale. The observed aerosol size distributions, chemical
composition and vertical velocity distribution (Gaussian with standard deviation $\sigma_w$) are introduced into a
state-of-the-art cloud droplet parameterization to show that cloud maximum supersaturations in the region
are low, ranging from 0.02 to 0.52% with an average of 0.14±0.05%. Based on these low values of
supersaturation, the majority of activated droplets correspond to particles of diameter 90 nm and above.
Droplet number shows little sensitivity to total aerosol owing to their strong competition for water vapor.
Given, however, that $\sigma_w$ exhibits considerable diurnal variability (ranging from 0.16 m s$^{-1}$ during nighttime
to over 1.2 m s$^{-1}$ during day), its covariance with total aerosol number ($N_a$) during the same period amplifies



predicted response in cloud droplet number ($N_d$) by 3 to 5 times. Therefore, correct consideration of vertical
velocity and its covariance with time and aerosol amount is important for fully understanding aerosol-cloud
interactions and the magnitude of the aerosol indirect effect. Datasets and analysis such as the one presented
here can provide the required constraints for addressing this important problem.

**1. Introduction**
Atmospheric particles (aerosols) interact with the incoming solar radiation through scattering and
absorption processes which tend to cool the Earth, especially over dark surfaces such as oceans and forests
(Brock et al., 2016a). Aerosols also act as cloud condensation nuclei (CCN) and subsequently form cloud
droplets and indirectly affect climate through modification of cloud radiative properties - an effect which
constitutes one of the most uncertain aspects of anthropogenic climate change (Seinfeld et al., 2016).
Studies often highlight the importance of constraining the aerosol size distribution, particle composition
and mixing state for predicting CCN concentrations (Cubison et al., 2008; Quinn et al., 2008). Model
assumptions often cannot consider the full complexity required to comprehensively compute CCN – which
together with other emissions and process uncertainties lead to CCN prediction errors that can be significant
(e.g., Fanourgakis et al., 2019). Owing to the sublinear response of cloud droplet number concentration ($N_d$)
to aerosol perturbations, prediction errors in CCN generally result in errors in $N_d$ which are less than those
for CCN (Fanourgakis et al., 2019). The sublinear response arises because elevated CCN concentration
generally increases the competition of the potential droplets for water vapor; this in turn depletes
supersaturation and the $N_d$ that can eventually form (Reutter et al., 2009; Bougiatioti et al., 2016;
Fanourgakis et al., 2019; Kalkavouras et al., 2019). A critically important parameter is the vertical velocity;
so important in fact that droplet number variability may be driven primarily by vertical velocity variations
(Kacarab et al., 2020; Sullivan et al., 2019). Compared to aerosols, vertical velocity is much less observed,
constrained and evaluated in aerosol-cloud interaction studies, hence may be a source of persistent biases
in models (Sullivan et al., 2019).
The Southeast United States (SEUS) presents a particularly interesting location for studying regional
climate change, as it has not considerably warmed over the past 100 years – except for the last decade
(Carlton et al., 2018; Yu et al., 2014; Leibensperger et al., 2012b). These trends are in contrast with the
trends observed in most locations globally (IPCC 2013), and several hypotheses have been proposed to
explain this regional phenomenon, including the effect of involving short-lived climate forcers such as
secondary aerosols combined with the enhanced humidity in the region and their impact on clouds (Carlton
et al., 2018; Yu et al., 2014). Here, we analyze data collected during the Southeast Nexus of Air Quality





and Climate (SENEX) campaign in June-July 2013, which was the airborne component led by the National
Oceanic and Atmospheric Administration (NOAA), of a greater measurement campaign throughout the
SEUS, the Southeast Atmosphere Study (SAS; Carlton et al., 2018). Here we analyze data collected onboard
the NOAA WP-3D and apply a state-of-the-art droplet parameterization to determine the maximum
supersaturation and $N_d$ achieved in cloudy updrafts, for all science flights with available number size
distribution and chemical composition data. We also determine the sensitivity of droplet formation to
vertical velocity and aerosol, with the purpose of understanding the drivers of droplet variability in the
boundary layer of the SEUS by obtaining regional-scale, representative values of the relationship between
the driving parameters and cloud droplet number.

**2. Methods**
*2.1 Aircraft instrumentation*
The analysis utilizes airborne, in situ data collected during the June-July 2013 SENEX mission, aboard the
National Oceanic and Atmospheric Administration (NOAA) WP-3D aircraft (typical airspeed ~100 m s$^{-1}$)
based in Smyrna, Tennessee (36°00'32''N, 86°31'12''W). In total, twenty research flights were conducted.
Based on the availability of the relevant data described below, thirteen flights are analyzed in this work.
Description of the analyzed research flights are provided in Table 1. Detailed information on the
instrumentation and measurement strategy during the SENEX campaign can be found in Warneke et al.

89   (2016).

Dry particle number distributions from 4 - 7000 nm were measured using multiple condensation and optical
particle counters. 4-700 nm particles were measured by a nucleation mode aerosol size spectrometer
(NMASS; Warneke et al., 2016) and an ultra-high sensitivity aerosol spectrometer (UHSA; Brock et al.,
2011), while for larger particles with dry diameters between 0.7 and 7.0 μm, a custom-built white-light
optical particle counter (WLOPC) was used.
Measurements of the composition of submicron vacuum aerodynamic diameter non-refractory aerosol (less
than 0.7 μm diameter) were made with a Compact Time-of-Flight Aerosol Mass Spectrometer (C-ToF-
AMS; Aerodyne, Billerica, Massachesetts, US) (Canagaratna et al., 2007; Kupc et al., 2018) customized
for aircraft use, with a 10 s time resolution (Warneke et al., 2016). Particles entering the instrument are
focused and impacted on a 600 °C inverted-cone vaporizer. The volatilized vapors are analyzed by electron
ionization mass spectrometry, providing mass loadings of sulfate, nitrate, organics, ammonium and
chloride. For the C-ToF-AMS, the transmission efficiency of particles between 100 and 700 nm is assumed
to be 100% through the specific aerodynamic focusing lens used while mass concentrations are calculated





using a chemical composition-dependent collection efficiency (Middlebrook et al., 2012; Wagner et al.,
2015). The C-ToF-AMS only measures non-refractory aerosol chemical composition, therefore this
analysis provides mass loadings of sulfate, nitrate, ammonium and organic constituents with a 10 s time
resolution and neglects the contribution of black carbon (BC). The calculation of the average volume
fractions from the mass loading follows that of Moore et al. (2012). An average organic density of 1.4 g
cm$^{-3}$ is used, characteristic of aged aerosol (Moore et al., 2011; Lathem et al., 2013) while for the inorganic
species the respective densities are used, assuming the aerosol to be internally mixed.
The aircraft was equipped by the NOAA Aircraft Operations Center (AOC) flight facility, incorporating a
suite of instruments to provide information on exact aircraft position as well as numerous meteorological
parameters (Warneke et al., 2016). The analysis in this work makes use of vertical wind velocity, aircraft
radar altitude, and ambient temperature, pressure and relative humidity (RH) provided by NOAA AOC.
Location of the instrumentation on the aircraft can be found elsewhere (Warneke et al., 2016). For
measurements inside the fuselage a low turbulence inlet (Wilson et al., 2004) and sampling system (Brock
et al., 2011; 2016a) was used to decelerate the sample flow to the instruments. The C-ToF-AMS was
connected downstream of an impactor with 50% efficiency at a 1.0 μm aerodynamic diameter (PM1) cut-
point (Warneke et al., 2016).
*2.2 Aerosol hygroscopicity parameter*
The aerosol hygroscopicity parameter (Petters and Kreidenweis, 2007), $\kappa$, is calculated assuming a mixture
of an organic and inorganic component with volume fraction $\varepsilon_{org}$, $\varepsilon_{inorg}$ and characteristic hygroscopicity
$\kappa_{org}$, $\kappa_{inorg,}$ respectively ($\kappa=\varepsilon_{inorg}\kappa_{inorg}+\varepsilon_{org}\kappa_{org}$). The organic and inorganic volume fraction are derived from
the C-ToF-AMS data. Since throughout the summertime SEUS, aerosol inorganic nitrate mass and volume
fraction are very low (Weber et al., 2016; Fry et al., 2018), $\kappa_{inorg}$ =0.6, representative for ammonium sulfate,
is used. For the organic fraction, a hygroscopicity value of $\kappa_{org}$=0.14 is used, based on concurrent
measurements conducted at the ground site of the SAS at the rural site of Centreville, Alabama (Cerully et
al., 2015). This value is also in accordance with the cumulative result of studies conducted in the Southeast
US using measurements of droplet activation diameters in subsaturated regimes, providing $\kappa_{org}$ of > 0.1
(Brock et al., 2016a).
*2.3 Cloud droplet number and maximum supersaturation*
Using the observed aerosol number size distribution (1 s time resolution) and the hygroscopicity derived
from the chemical composition measurements (10 s time resolution), we calculate the droplet number ($N_d$)
and maximum supersaturation ($S_{max}$) that would form in clouds in the airmasses sampled. Droplet number



and maximum supersaturation calculations are carried out at a regional scale using an approach similar to
that of Bougiatioti et al. (2016) and Kalkavouras et al. (2019) with the sectional parameterization of Nenes
and Seinfeld (2003), later improved by Barahona et al. (2010) and Morales Betancourt and Nenes (2014a).
A sectional representation of the size distribution is used and provided for each data point of each flight
(per second, e.g. for Flight 5, n=23213 data points). Given that chemical composition is provided with a 10
s time resolution, the same hygroscopicity values are used for 10 size distributions during each flight.
Temperature and pressure required for droplet number calculations are obtained from the NOAA AOC
flight facility dataset.
Droplets form from activation of aerosol in cloudy updrafts, so here we use the available measurements of
vertical velocity together with the aerosol measurements to derive a potential cloud droplet concentration.
Given that vertical velocity varies considerably inside the boundary layer we represent droplet number with
the average concentration that results from integrating over the distribution (probability density function,
PDF) of observed updraft velocities. To accomplish this, each flight is divided in segments where the
aircraft flew at a constant height. For each segment, the positive vertical velocities are fit to a Gaussian
distribution with mean of zero and width of spectral dispersion $\sigma_w$. Positive vertical velocities ("updrafts")
were used, as they are the part of the vertical velocity spectrum that is responsible for cloud droplet
formation. The $\sigma_w$ values derived from the level leg segments are then averaged into one single $\sigma_w$ value
(and standard deviation) to represent the flight. Application of the "characteristic velocity" approach
(Morales and Nenes, 2010) then gives the PDF-averaged droplet number concentration by calling the
droplet parameterization at a single "characteristic" velocity, $w^*=0.79\sigma_w$ (Morales and Nenes, 2010). This
calculation approach is applied to each size distribution measured. Apart from its theoretical basis, this
methodology has shown to provide closure with observed droplet numbers in ambient clouds (e.g. Kacarab
et al., 2020).
In determining $\sigma_w$, we consider segments that are expected to be in the boundary layer: 91 % of the segments
are below 1000 m (mean altitude ~700 m; Table 2) typically corresponding to the height of the boundary
layer in the summertime US (Seidel et al., 2013). The vertical velocity distributions observed gave $\sigma_w$
=0.97±0.21 m s$^{-1}$ for daytime flights, and $\sigma_w$ =0.23±0.04 m s$^{-1}$ for nighttime flights (Table 2). Because of
this strong diurnal variation in $\sigma_w$, potential droplet formation is evaluated at four vertical velocities that
cover the observed ranged, namely 0.1, 0.3, 0.6 and 1 m s$^{-1}$.
We also compute the sensitivity of the derived $N_d$, to changes in aerosol number concentration ($N_a$), $\kappa$ and
$\sigma_w$, expressed by the partial derivatives $\partial N_d/\partial N_a$, $\partial N_d/\partial \kappa$ and $\partial N_d/\partial \sigma_w$ computed from the parameterization
using a finite difference approximation (Bougiatioti et al., 2016; Kalkavouras et al., 2019). These
sensitivities, together with the observed variance in $N_a$, $\kappa$, and $\sigma_w$ are also used to attribute droplet number



variability to variations in the respective aerosol and vertical velocity parameters following the approach
of Bougiatioti et al. (2016) and Kalkavouras et al. (2019).

**3. Results and Discussion**
*3.1. Particle composition and size distribution*
For the determination of the different aerosol species present, neutral and acidic sulfate species are
distinguished by the molar ratio of ammonium to sulfate ions. A molar ratio higher than 2 indicates the
presence of only ammonium sulfate, while values between 1 and 2 indicate the presence of both ammonium
sulfate and bisulfate (Seinfeld and Pandis, 1998). For most of the flights, the molar ratio of ammonium
versus sulfate was well above 2, having a mean value of 2.41±0.72 (median 2.06). For the nighttime flights
the values were somewhat lower (1.91±0.42 and median of 1.85, respectively). Nevertheless, ammonium
sulfate is always the predominant sulfate salt. Organic mass fractions for the SENEX research flights are
provided in Table 1. Overall, organic aerosol was found to dominate during all flights, contributing 66%-
75% of the total aerosol volume. Most of the remaining aerosol volume consists of ammonium sulfate,
ranging from 12%-39% (with a mean of 23%±6%). The organic mass fraction during the flights was found
to decrease with height (see Fig. 1). This vertical variability of the chemical composition can have a strong
impact on droplet number within the boundary layer, as air masses from aloft may descend and interact
with that underneath. Figure 1 represents the organic mass fractions during Flights 5 and 12. The lowest
organic mass fractions overall were observed during Flight 12 (35%±18% with values < 5% for altitudes
>3000 m, Fig.1b) while the highest ones were observed at flights over predominantly rural areas (Flights 5
(Fig. 1c) , 10 and 16). During Flight 5 the organic mass fraction was high (68%±5%), with the highest
values found in the free troposphere at altitudes > 3000 m. High organic mass fractions were also found
during a nighttime flight that included portions of the Atlanta metropolitan area, with values up to 78%.
The impact of the chemical variability on droplet number is discussed in section 3.2.
The predominance of the organic fraction is also reflected in the hygroscopicity parameter values, with an
overall $\kappa = 0.25±0.05$, which is close to the proposed global average of 0.3 (Seinfeld and Pandis, 1998).
The highest values are, as expected, for flights exhibiting the lowest organic mass fraction, namely Flight
12 with a $\kappa = 0.39$ (Table 2). The rest of the κ-values are close to the overall values, as the organic mass
fractions are around 0.65.
Median aerosol size distributions are obtained from the median and interquartile range in each size bin from
the aerosol size distribution measurements during segments where the aircraft flew at a constant height.
The impact of the variability of the total aerosol number on droplet number is discussed in detail further in





section 3.2. Overall, number concentrations ranged from around 500 to over 100000 cm$^{-3}$ with number size
distributions varying markedly over the course of a flight. In general, free tropospheric distributions
exhibited characteristics of a bimodal distribution with a prominent broad accumulation mode peak (80-
200 nm) and an Aitken mode peak (30-60 nm) (Fig. 2a) while boundary layer size distributions exhibited a
more prominent accumulation mode (Fig. 2b). There was considerable variability in the contributions of
the nucleation, Aitken, and accumulation modes to the total aerosol number, depending on altitude and
proximity to aerosol sources (Fig. 2c). Nevertheless, the modal diameters did not vary much. Distributions
during nighttime flights exhibited similar total aerosol number and variability; nevertheless, size
distributions were more complex exhibiting even three different modes (20-40, 70-100 and 130-200 nm;
Fig. 2d). Considering that mostly particles in the accumulation mode activate into cloud droplets (particles
with diameters >90 nm), contrasts between day and nighttime aerosol characteristics/variability may not be
as large, and driven primarily by the total aerosol number in the accumulation mode.
*3.2 Potential cloud droplet number*
The calculation of $N_d$ and $S_{max}$, was carried out for all thirteen research flights. Results are given in Tab. 3
for the four different values (0.1, 0.3, 0.6 and 1 m s$^{-1}$) of $\sigma_w$. Overall it can be seen that for all flight
conditions and for low $\sigma_w$, $N_d$ shows a low variance (mean of 132±20 for 0.1 m s$^{-1}$ and 350±100 for 0.3 m
s$^{-1}$). For a given $\sigma_w$, the variance of $N_d$ is predominantly attributed to relative changes in $N_a$ rather than
changes in the chemical composition (expressed by changes in the hygroscopicity parameter, $\kappa$). The
highest relative contribution of the chemical composition (12% and 35% for 0.1 and 0.3 m s$^{-1}$, respectively)
to the variation of $N_d$ is found for Flight 18, during which the total aerosol number was the lowest. Indicative
of a "cleaner" environment; the organic mass fraction was relatively lower and the hygroscopicity
parameter was higher. Even though the lowest organics mass fraction and highest $\kappa$ were observed during
Flight 12, droplet formation is much more sensitive to changes in aerosol concentration than to variations
in composition.
As the vertical velocity increases, so does supersaturation and consequently the droplet number (by 62%
from 0.1 to 0.3 m s$^{-1}$, 70% from 0.3 to 0.6 m s$^{-1}$ and another 39% from 0.6 to 1 m s$^{-1}$). The relative
contribution of the chemical composition to the variation of cloud droplet number increases from 5±3% for
0.1 m s$^{-1}$, to 12.3±8% for 0.3 m s$^{-1}$, to 14.5±10% for 0.3 m s$^{-1}$ and 16.5±9% for 1 m s$^{-1}$. The highest droplet
numbers are estimated for Flights 6 and 10, which included urban environments during daytime (Atlanta).
Overall during daytime, when $\sigma_w$ varies little and is large, and $N_a$ is high, the relative contribution of $N_a$ to
the variation of $N_d$ is the highest (more than 90%) while the relative contribution of $\kappa$ is limited (less than
10%) (see Table 3, Flights 10, 11, 12, 17 and 19). Turbulence is limited during nighttime when $\sigma_w$ is the



lowest (0.23±0.04); therefore, the $\sigma_w = 0.3$ m s$^{-1}$ case is most representative of nighttime conditions. During
daytime, when $\sigma_w$ is high (0.97±0.21), $\sigma_w = 1$ m s$^{-1}$ should be considered as most representative.
As $\sigma_w$ varies considerably throughout the day, we estimate its contribution together with variations in $N_a$
and $\kappa$, to the total variability in $N_d$ based on $\partial N_d/\partial \kappa$, $\partial N_d/\partial N_a$ and $\partial N_d/\partial \sigma_w$ and the variances of $\kappa$, $N_a$ and $\sigma_w$
(Table 4). The $\sigma_w$ variation during nighttime, although small (always less than 10%), consistently remains
an important contributor to $N_d$ variability, because droplet formation tends to be in the updraft velocity-
limited regime. At higher values of $\sigma_w$ (Table 4), the contribution of $\sigma_w$ variability to $\underline{N_d}$ variability is reduced
and dominated by $N_a$ variability.
To explore the importance of aerosol compared to updraft velocity, we focus on two pairs of flights
conducted in two sectors, from each sector one during day- and one during night-time (see Fig.3). In both
pairs of flights (Flight 5 and 15, and Flight 6 and 9), $\sigma_w$ varies about the same between night and day (Table
4). For the first pair of flights, the daytime variability in $N_d$ (which is 69%) is to within 75% driven by
aerosol (69% by $N_a$ and 7% from $\kappa$) and 24% by $\sigma_w$. For nighttime, 58% by aerosol (51% by $N_a$ and 7%
from $\kappa$) and 42% of the variability is driven by $\sigma_w$. For the second pair of night/day flights, $N_a$ is on average
similar, $\sigma_w$ varies by a factor of 4.0 and $\kappa$ varies by 13%. Attribution calculations suggest that the diurnal
variability in $N_d$ (where daytime values are 72.1% higher than nighttime) is 3, 54 and 43% for $\kappa$, $N_a$ and $\sigma_w$,
respectively during day and 7, 76, and 17% driven by $\kappa$, $N_a$ and $\sigma_w$, respectively during night (Table 4). In
the second sector, 57% of the variability in $N_d$ is driven by aerosol during the day and 83% during the night.
As expected, droplet number ($N_d$) and maximum supersaturation ($S_{max}$) increases as $\sigma_w$ becomes larger. The
highest $S_{max}$ are around 0.2-0.3% and found for flights which exhibited large and highly variable $\sigma_w$ (Flights
4, 5, 12 and 19) while the lowest $S_{max}$ are around 0.10% and found for the nighttime flights (Flights 9, 15
and 16). All other flights yield similar $S_{max}$, which are around 0.13%. Based on the calculated $S_{max}$ for every
flight, the majority of the activated droplets correspond to particles of 90 nm diameter and above. Figure 3
presents the calculated $N_d$ for the four aforementioned flights, namely Fights 5 (Fig. 3a), 15 (Fig. 3b), 6
(Fig. 3c) and 9 (Fig. 3d) using the observed $\sigma_w$. The size of the markers represents the number of droplets,
while the color scale the respective total aerosol number.
Figure 4 shows $N_d$ relative to $N_a$ for flights conducted in two sectors, during day and night (Flights 5 & 15,
and Flights 6 & 9, respectively). It can be seen that throughout these flights, $N_d$ reaches a plateau, where
any additional aerosol does not translate to any significant increase in $N_d$. This plateau is caused by strong
water vapor limitations and is different for day and night. $S_{max}$ is lower during night because vertical wind
velocity, ambient T and RH are lower. The same factors cause that for Flight 6 & 9 (Fig. 4c & d) where $N_a$
was almost the same, $N_d$ is almost 3.5 times lower during night (Flight 9). For the whole dataset (13 flights),





results are summarized in Figure 5, where droplet numbers are calculated based on the observed $\sigma_w$ and the
respective "characteristic", mean velocities, $w^*$. Under low $w^*$ conditions, $N_a$ variability does not result in
an important change in $N_d$. On the contrary, when $w^*$ tends to increase and $N_a$ increases, as is characteristic
of polluted regions, during daytime, then the impact on droplet number is more notable. This point is evident
in Figure 6, comprising the different segments of the flights when the aircraft sampled at practically the
same altitude within the boundary layer. It can be seen that $N_a$ is enhanced as $w^*$ increases. The lowest $w^*$
values (shaded area) correspond to the segments of the flights during nighttime.
Overall, $S_{max}$ of clouds from all the evaluated SENEX data, is 0.14±0.05%. Tripling $\sigma_w$ from 0.1 to 0.3 m s⁻
¹ results in 31% increase in $S_{max}$, while doubling from 0.3 to 0.6 m s⁻¹ results in 26.2% increase in $S_{max}$ and
a further $\sigma_w$ increase to 1 m s⁻¹ leads to an additional 20.7% increase in $S_{max}$. Overall effect of updraft
velocity on calculated $N_d$: tripling $\sigma_w$ from 0.1 to 0.3 m s⁻¹ results in a 61.9% increase in $N_d$, doubling from
0.3 to 0.6 m s⁻¹ results in 40.5% $N_d$ increase; increasing $\sigma_w$ to 1 m s⁻¹ leads to an additional 26.9% increase
in $N_d$. Furthermore, for a given $\sigma_w$, despite of the presence or not of a large number of aerosol (e.g. Flight
10 where $N_a$ is 2.7 times higher than $N_a$ in Flight 15) the difference in calculated $N_d$ for 0.6 m s⁻¹ is only 1.3
times higher for Flight 10 than Flight 15. This highlights the relative insensitivity of $N_d$ to variations in $N_a$
for constant $\sigma_w$.

**4. Summary and Conclusions**
Measurements of wind velocity, ambient conditions (*T, RH*), aerosol number size distribution and
composition in the SEUS obtained during the SENEX 2013 project are used to analyze the drivers of droplet
formation. Overall 13 research flights are studied, covering environments over sectors with different aerosol
sources, impacting total aerosol number, size distribution and chemical composition. Aerosol volume is
largely dominated by an organic fraction resulting in a calculated hygroscopicity of 0.25±0.05.
Based on the calculation of cloud droplet number concentration ($N_d$) and maximum supersaturation ($S_{max}$),
we find that on a regional scale, most of the variability of $N_d$ is due to the fluctuations in $N_a$ (Table 4), in
accordance with other recent studies (Fanourgakis et al., 2019). Nonetheless, $N_d$ levels are also sensitive to
fluctuations in $\sigma_w$, as a variation by a factor of 4.0 in $\sigma_w$ may lead to an $N_d$ variation of almost a factor of
3.6 and at the same time the $N_d$ response to different $N_a$ levels may be enhanced by a factor of 5 (Figure 4).
$S_{max}$ changes in response to aerosol concentration, in a way that tends to partially mitigate $N_d$ responses to
aerosol. Overall, maximum supersaturation levels remain quite low (0.14±0.05%) with predicted levels
being much lower in lower altitudes (0.05±0.1%). Because of the strong competition for water vapor
(expressed by the low $S_{max}$), cloud droplet number exhibits enhanced sensitivity to aerosol number





variations throughout the flights, regardless of aerosol composition. On the other hand, droplet
concentration especially within the boundary layer approaches a "plateau" that is strongly driven by vertical
velocity (turbulence) and the resulting supersaturation, but also aerosol concentration. In "cleaner"
environments where total aerosol number is lower, the relative contribution of vertical velocity to cloud
droplet number is almost half during nighttime (24% vs. 42% during daytime) while the relative
contribution of $N_a$ to the variance in $N_d$ is somewhat higher (69% vs. 51% during daytime) even though $N_a$
is 2-fold lower during night. On the contrary, in environments with elevated concentrations of
accumulation-mode particles, the majority of cloud droplet number variations (54% during nighttime vs.
76% during daytime) can be attributed to changes in total aerosol number and to a lesser extent to vertical
velocity (43% during nighttime vs. 17% during daytime). The relative contribution of the total aerosol
number to the cloud droplet number dominates over variations in chemical composition (expressed by $\kappa$).
There are cases however where chemical composition variability contributes a non-negligible (~9%)
contribution to droplet number variability.
Overall, our results show that atmospheric dynamics is a key driver of cloud droplet formation and its
variability in the region. Especially in cases when the boundary layer turbulence is low (e.g. during
nighttime), low vertical velocity, generating only small supersaturation, can be as important a contributor
to droplet number variability as aerosol number. For cases with high vertical velocities and high aerosol
number concentration, it is the aerosol concentration that dominates the variability in cloud droplet number.
On average, the two variables ($N_a$ and $\sigma_w$) contribute almost equally to the variability in cloud droplet
number concentration ($N_d$) and together account for more than 90% of variability. This finding is consistent
with recent modeling studies noting the importance of vertical velocity variability as a driver of the temporal
variability of global hydrometeor concentration (Morales Betancourt and Nenes, 2014b; Sullivan et al.,
2016). Furthermore, the $N_d$ enhancement from changes in $N_a$ is magnified up to 5 times from concurrent
changes in $\sigma_w$. A similar situation has also been observed in smoke-influenced marine boundary layers in
the S.Atlantic (Kacarab et al., 2020). Altogether, these findings carry important implications for model
assessments of aerosol indirect climate forcing (e.g., Leibensperger et al., 2012a) and aerosol-cloud
interaction studies using remote sensing, as patterns of cooling (although consistent with aerosol and cloud
fields) may omit the covariance of vertical velocity with aerosol number, therefore neglecting this important
driver of hydrometeor variability.
**Data Availability:** The data used in this study can be downloaded from the NOAA public data repository
at https://www.esrl.noaa.gov/csd/projects/senex/. The Gaussian fits used for determining $\sigma_w$ and the droplet
parameterization used for the calculations in the study are available from athanasios.nenes@epfl.ch upon
request.



**Author Contributions:** conceptualization, A.B. and A.N.; methodology, A.B. and A.N.; software, A.N.; formal analysis, A.B. and A.N.; investigation, A.B., A.N. and J.J.L.; data curation, A.B., J.J.L., C.B., J.A.G., J.L., A.M.M., A.W.; writing—original draft preparation, A.B. and A.N.; writing—review and editing, A.B., A.N., J.J.L., C.B., A.W., additional comments by A.M.M.; visualization, A.B. A.N. and J.J.L.; supervision, A.N.; project administration, A.N.; funding acquisition, A.N.

**Funding:** This study was supported by the Environmental Protection Agency STAR Grant R835410, the Action "Supporting of Postdoctoral Researchers" of the Operational Program "Education and Lifelong Learning" (action's beneficiary: General Secretariat for Research and Technology) and is co-financed by the European Social Fund (ESF) and the Greek State. We also acknowledge funding from the European Research Council, CoG-2016 project PyroTRACH (726165) funded by H2020-EU.1.1. – Excellent Science.

**Conflicts of Interest:** The authors declare no conflict of interest.

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



**Table 1:** Research flights from SENEX 2013 used in this study.

| Flight | Date | Local Time (CDT, UTC-5 hrs) | $\kappa$ | Organic mass fraction |
|---|---|---|---|---|
| **4** | 10/6 | 09:55-16:30 | 0.23±0.02 | 0.62±0.11 |
| **5** | 11/6 | 11:30-17:57 | 0.20±0.00 | 0.68±0.05 |
| **6** | 12/6 | 09:48-15:31 | 0.21±0.01 | 0.68±0.07 |
| **9** | 19/6 | 17:30-23:29 | 0.24±0.01 | 0.66±0.06 |
| **10** | 22/6 | 10:01-17:09 | 0.21±0.02 | 0.68±0.08 |
| **11** | 23/6 | 10:08-17:22 | 0.25±0.03 | 0.58±0.07 |
| **12** | 25/6 | 10:18-17:25 | 0.39±0.02 | 0.35±0.18 |
| **14** | 29/6 | 10:26-17:39 | 0.22±0.03 | 0.62±0.07 |
| **15** | 2/7 | 20:08-02:51 | 0.28±0.05 | 0.55±0.09 |
| **16** | 3/7 | 19:56-02:55 | 0.22±0.05 | 0.67±0.09 |
| **17** | 5/7 | 09:52-16:24 | 0.23±0.05 | 0.59±0.14 |
| **18** | 6/7 | 09:19-16:18 | 0.31±0.02 | 0.52±0.08 |
| **19** | 8/7 | 10:11-16:44 | 0.23±0.04 | 0.62±0.08 |







**Table 2:** Flight number, time interval, spectral dispersion of vertical wind velocity ($\sigma_w$) and characteristic
vertical velocity $w^*=0.79\sigma_w$ during flight segments where the aircraft flew at a constant altitude.

| Flight (pass) | Time Range | $\sigma_w$ (m s$^{-1}$) | $w^*$ (m s$^{-1}$) | Altitude (m) | Flight (pass) | Time Range | $\sigma_w$ (m s$^{-1}$) | $w^*$ (m s$^{-1}$) | Altitude (m) |
|---|---|---|---|---|---|---|---|---|---|
| 5 (1) | 12:31-12:58 | 1.02 | 0.81 | 549± 58 | 9 (1) | 18:44-18:58 | 0.255 | 0.202 | 797±2.01 |
| 5 (2) | 13:16-13:29 | 0.82 | 0.65 | 982±11 | 9 (2) | 19:20-19:29 | 0.249 | 0.197 | 740±1.23 |
| 5 (3) | 13:34-13:50 | 1.01 | 0.80 | 502±13 | 9 (3) | 19:33-19:48 | 0.217 | 0.171 | 740±1.23 |
| 5 (4) | 13:53-14:08 | 1.03 | 0.81 | 614±27 | 9 (4) | 19:51-20:25 | 0.218 | 0.173 | 776±1.22 |
| 5 (5) | 14:20-15:00 | 0.91 | 0.72 | 603±40 | 9 (5) | 20:34-20:39 | 0.232 | 0.183 | 597±1.19 |
| 5 (6) | 15:35-15:41 | 0.87 | 0.69 | 533±18 | 9 (7) | 20:56-21:10 | 0.201 | 0.158 | 773±1.11 |
| 5 (7) | 16:17-16:30 | 0.77 | 0.61 | 638±23 | 9 (8) | 21:31-21:45 | 0.191 | 0.151 | 725±1.18 |
| 5 (8) | 16:31-16:39 | 0.55 | 0.44 | 559±18 | 9 (9) | 22:24-22:31 | 0.257 | 0.203 | 745± 1.36 |
| 5 (9) | 17:10-17:22 | 0.53 | 0.42 | 686±40 | 9 (10) | 22:48-22:54 | 0.221 | 0.175 | 804± 1.37 |
| 14 (1) | 12:34-12:49 | 0.94 | 0.75 | 558±2 | 15 (1) | 21:09-21:52 | 0.236 | 0.186 | 505±6.64 |
| 14 (2) | 13:57-14:17 | 0.97 | 0.77 | 658±3 | 15 (2) | 22:19-22:31 | 0.301 | 0.238 | 633±1.21 |
| 14 (3) | 14:22-14:46 | 0.95 | 0.75 | 737±3 | 15 (3) | 22:42-22:54 | 0.255 | 0.202 | 600±1.17 |
| 14 (4) | 14:58-15:33 | 0.55 | 0.43 | 746±23 | 15 (4) | 23:26-23:37 | 0.329 | 0.260 | 908±1.56 |
| 14 (5) | 15:55-16:08 | 0.57 | 0.45 | 714±3 | 15 (5) | 00:02-00:19 | 0.297 | 0.235 | 1208±1.23 |
| 14 (6) | 16:11-16:21 | 0.77 | 0.61 | 801±3 | 15 (6) | 00:43-1:08 | 0.253 | 0.199 | 592±1.37 |
| 14 (7) | 16:33-16:41 | 0.45 | 0.35 | 793± 2 | 15 (7) | 1:10-1:24 | 0.276 | 0.218 | 676±1.02 |
|  |  |  |  |  | 15 (8) | 1:37-2:02 | 0.207 | 0.164 | 713±19.5 |
| 12 (1) | 11:50-12:34 | 0.96 | 0.75 | 484±3 | 19 (1) | 11:20-11:41 | 0.622 | 0.492 | 1014±2.27 |
| 12 (2) | 12:48-13:18 | 1.09 | 0.86 | 503±3 | 19 (2) | 12:09-12:23 | 1.203 | 0.95 | 652±3.34 |
| 12 (3) | 13:34-13:50 | 1.12 | 0.88 | 894±3 | 19 (3) | 12:51-13:10 | 0.873 | 0.689 | 537±2.51 |
| 12 (4) | 14:06-14:40 | 1.04 | 0.82 | 479±4 | 19 (4) | 13:22-13:49 | 1.294 | 1.022 | 518±22.6 |
| 12 (5) | 15:21-15:32 | 1.10 | 0.87 | 521±3 | 19 (5) | 14:44-14:57 | 1.361 | 1.075 | 528±3.26 |
| 12 (6) | 15:43-16:02 | 0.99 | 0.78 | 475±3 | 19 (6) | 15:04-16:06 | 0.896 | 0.708 | 524±2.8 |





**Table 3:** Derived cloud parameters (maximum supersaturation, droplet number) and relative contribution of chemical composition and total aerosol number for different vertical velocities. Numbers in parentheses indicate standard deviation values.

| Flight | $N_a$ | $N_a$ variab | $\sigma_w=0.1$ m s$^{-1}$ | | | | $\sigma_w=0.3$ m s$^{-1}$ | | | | $\sigma_w=0.6$ m s$^{-1}$ | | | | $\sigma_w=1.0$ m s$^{-1}$ | | | |
|---|---|---|---|---|---|---|---|---|---|---|---|---|---|---|---|---|---|---|
| | | | $S_{max}$ | $N_d$ | Cont $\kappa$ | Cont $N_a$ | $S_{max}$ | $N_d$ | Cont $\kappa$ | Cont $N_a$ | $S_{max}$ | $N_d$ | Cont $\kappa$ | Cont $N_a$ | $S_{max}$ | $N_d$ | Cont $\kappa$ | Cont $N_a$ |
| 4 | 6118 | 4520 | 0.11 (0.06) | 122 (41) | 0.08 | 0.92 | 0.16 (0.09) | 315 (114) | 0.20 | 0.80 | 0.21 (0.12) | 520 (212) | 0.23 | 0.77 | 0.26 (0.17) | 737 (321) | 0.2 | 0.8 |
| 5 | 4324 | 2598 | 0.08 (0.04) | 139 (31) | 0.09 | 0.91 | 0.1 (0.06) | 388 (104) | 0.15 | 0.85 | 0.14 (0.08) | 712 (216) | 0.17 | 0.83 | 0.17 (0.1) | 1063 (360) | 0.21 | 0.79 |
| 6 | 4958 | 3054 | 0.07 (0.07) | 151 (24) | 0.03 | 0.97 | 0.08 (0.04) | 422 (70) | 0.11 | 0.89 | 0.1 (0.06) | 773 (171) | 0.08 | 0.92 | 0.13 (0.07) | 1162 (302) | 0.07 | 0.93 |
| 9 | 4271 | 3095 | 0.07 (0.02) | 152 (18) | 0.05 | 0.95 | 0.12 (0.04) | 367 (68) | 0.17 | 0.83 | 0.16 (0.05) | 533 (115) | 0.17 | 0.83 | 0.19 (0.06) | 680 (126) | 0.12 | 0.88 |
| 10 | 6286 | 7201 | 0.07 (0.03) | 158 (24) | 0.02 | 0.98 | 0.1 (0.05) | 422 (86) | 0.02 | 0.98 | 0.14 (0.07) | 748 (180) | 0.04 | 0.96 | 0.18 (0.08) | 1063 (295) | 0.09 | 0.91 |
| 11 | 5969 | 7271 | 0.04 (0.01) | 137 (19) | 0.01 | 0.99 | 0.06 (0.01) | 381 (61) | 0.04 | 0.96 | 0.08 (0.02) | 695 (134) | 0.03 | 0.97 | 0.10 (0.02) | 1025 (226) | 0.03 | 0.97 |
| 12 | 3154 | 5150 | 0.06 (0.03) | 110 (45) | 0.03 | 0.97 | 0.1 (0.04) | 274 (117) | 0.05 | 0.95 | 0.14 (0.04) | 404 (179) | 0.08 | 0.92 | 0.17 (0.05) | 486 (207) | 0.07 | 0.93 |
| 14 | 5564 | 5891 | 0.07 (0.02) | 118 (41) | 0.05 | 0.95 | 0.10 (0.03) | 328 (125) | 0.17 | 0.83 | 0.13 (0.04) | 590 (240) | 0.25 | 0.75 | 0.16 (0.05) | 842 (361) | 0.27 | 0.73 |
| 15 | 2328 | 1428 | 0.05 (0.01) | 135 (22) | 0.03 | 0.97 | 0.09 (0.02) | 339 (67) | 0.12 | 0.88 | 0.12 (0.02) | 557 (137) | 0.21 | 0.79 | 0.16 (0.03) | 717 (203) | 0.3 | 0.7 |
| 16 | 3440 | 4507 | 0.08 (0.06) | 158 (37) | 0.03 | 0.97 | 0.12 (0.1) | 403 (120) | 0.06 | 0.94 | 0.17 (0.13) | 670 (235) | 0.07 | 0.93 | 0.23 (0.16) | 917 (374) | 0.1 | 0.9 |
| 17 | 3813 | 4645 | 0.05 (0.02) | 129 (41) | 0.06 | 0.94 | 0.07 (0.03) | 342 (130) | 0.1 | 0.9 | 0.1 (0.04) | 593 (248) | 0.06 | 0.94 | 0.13 (0.05) | 841 (371) | 0.06 | 0.94 |
| 18 | 1925 | 983 | 0.08 (0.04) | 90 (58) | 0.12 | 0.88 | 0.12 (0.05) | 233 (157) | 0.35 | 0.65 | 0.15 (0.06) | 379 (262) | 0.37 | 0.63 | 0.19 (0.07) | 499 (346) | 0.27 | 0.73 |
| 19 | 4323 | 7261 | 0.06 (0.02) | 121 (33) | 0.02 | 0.98 | 0.08 (0.02) | 314 (96) | 0.06 | 0.94 | 0.12 (0.03) | 526 (177) | 0.11 | 0.89 | 0.15 (0.03) | 670 (249) | 0.13 | 0.87 |



**Table 4:** Derived $S_{max}$, $N_d$, $\sigma_w$ for all research flights along with the estimated contribution of each
parameter to the variability of the droplet number.

| Flight | $\sigma_w$ (m s$^{-1}$) | $\dfrac{\Delta\sigma_w}{\sigma_w}$ | $S_{max}$ (%) | $N_d$ (cm$^{-3}$) | $\dfrac{\Delta N_d}{N_d}$ | Contrib. $\kappa$ | Contrib. $N_a$ | Contrib. $\sigma_w$ |
|---|---|---|---|---|---|---|---|---|
| 4 | 1.03±0.25 | 0.243 | 0.29±0.19 | 707±343 | 0.485 | 4% | 79% | 17% |
| 5 | 0.97±0.1 | 0.103 | 0.17±0.10 | 1040±350 | 0.337 | 7% | 69% | 24% |
| 6 | 0.94±0.18 | 0.191 | 0.13±0.07 | 1108±283 | 0.255 | 3% | 54% | 43% |
| 9 | 0.23±0.02 | 0.043 | 0.10±0.03 | 309±51 | 0.165 | 7% | 76% | 17% |
| 10 | 1.22±0.11 | 0.090 | 0.12±0.03 | 1177±271 | 0.230 | 1% | 90% | 9% |
| 11 | 1.08±0.04 | 0.037 | 0.11±0.03 | 1082±242 | 0.224 | 1% | 83% | 16% |
| 12 | 1.05±0.07 | 0.067 | 0.18±0.05 | 495±210 | 0.424 | 2% | 96% | 2% |
| 14 | 0.85±0.2 | 0.024 | 0.15±0.04 | 761±321 | 0.422 | 9% | 72% | 19% |
| 15 | 0.28±0.04 | 0.143 | 0.08±0.02 | 321±63 | 0.196 | 7% | 51% | 42% |
| 16 | 0.20±0.04 | 0.200 | 0.10±0.08 | 289±79 | 0.273 | 2% | 65% | 33% |
| 17 | 0.71±0.26 | 0.366 | 0.15±0.11 | 742±280 | 0.377 | 1% | 71% | 28% |
| 18 | 0.90±0.06 | 0.067 | 0.31±0.18 | 538±325 | 0.604 | 7% | 83% | 10% |
| 19 | 0.99±0.31 | 0.313 | 0.15±0.03 | 699±248 | 0.355 | 4% | 88% | 8% |

516





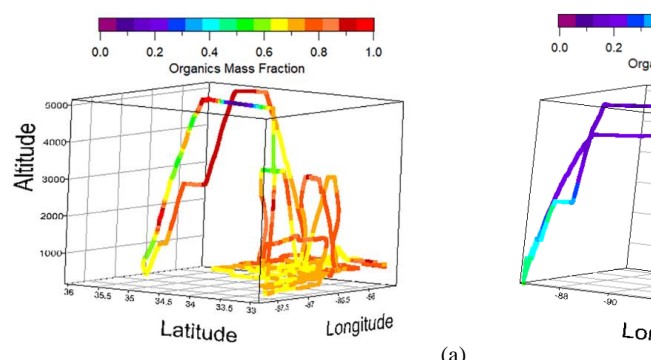

(a)                                                                              (b)

**Figure 1:** Spatial and vertical distribution of the organics mass fraction (a) for Flight 5 and (b) for Flight 12, denoting the difference in chemical composition, which in turn, may influence cloud droplet number concentration. The color scale denotes the percentage of the organics mass fraction.



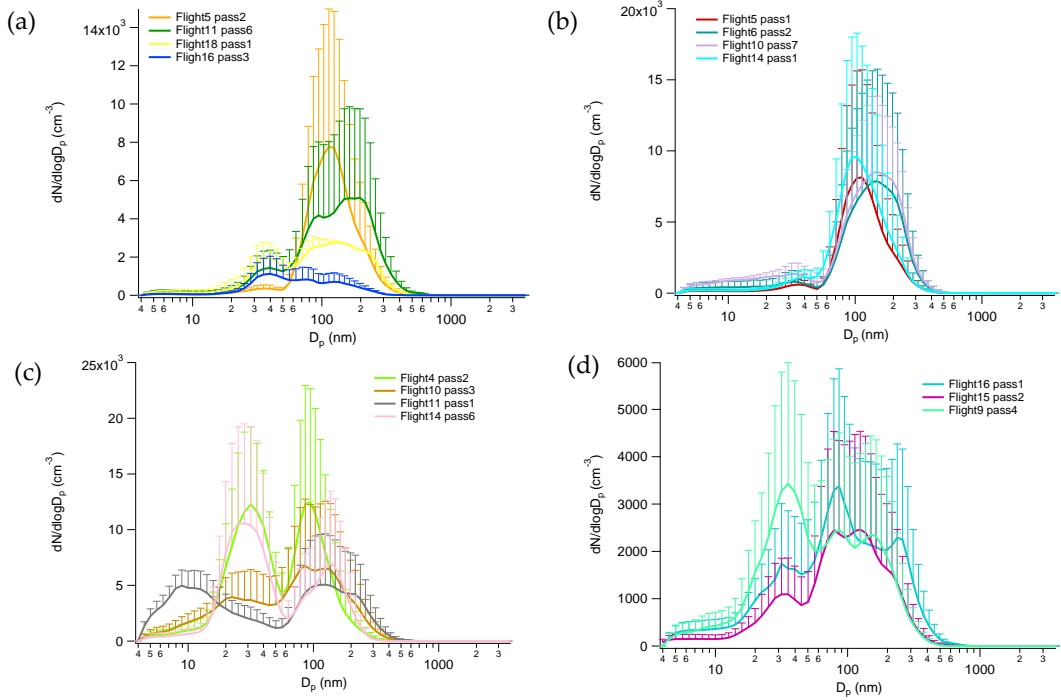

521

**Figure 2:** Average particle number size distributions for: (a) free tropospheric conditions, (b) within the
boundary layer, (c) for flights with high variability in total aerosol number, and (d) during nighttime flights.
Error bars represent the 75[th] percentile of the distributions within each pass.

525

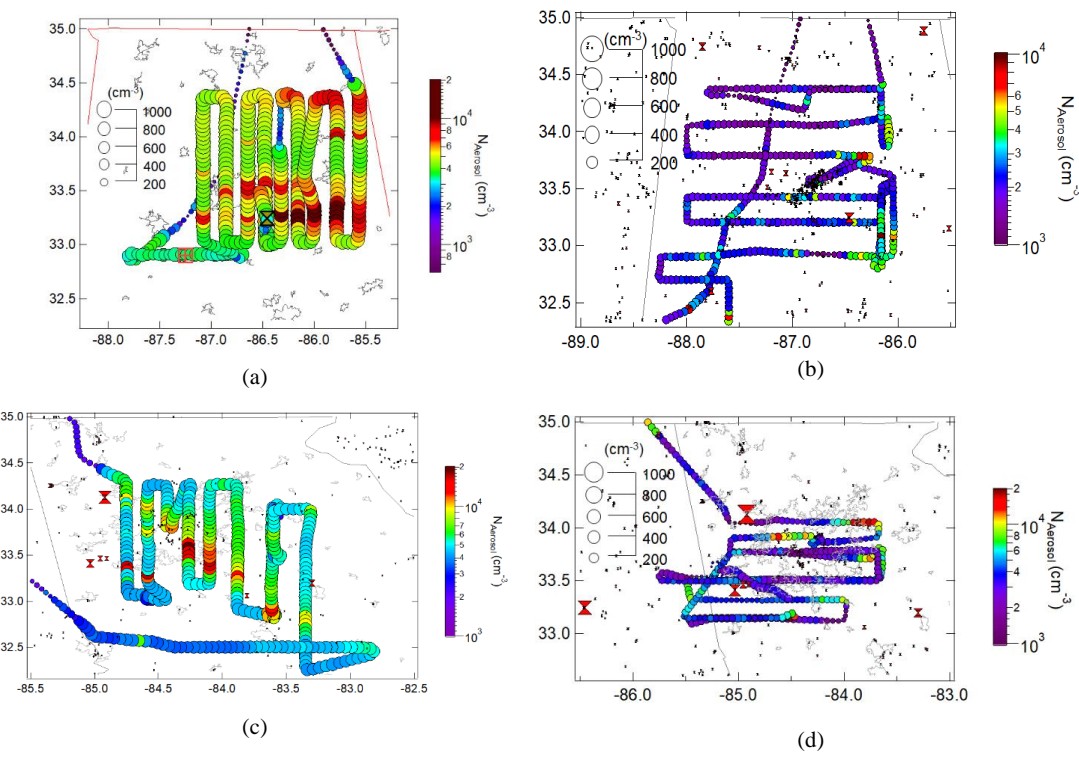

526

**Figure 3:** Flight trajectories showing cloud droplet number (indicated by marker size (cm⁻³)) and total
aerosol number (indicated by marker color) for the observed characteristic vertical velocity ($w*$). (a) for the
Alabama sector during daytime (Flight 5) and (b) nighttime (Flight 15). (c) for Atlanta during daytime
(Flight 6) and (d) nighttime (Flight 9). Note that the data are plotted at less than 1 Hz in order to better show
the size and color of the markers.






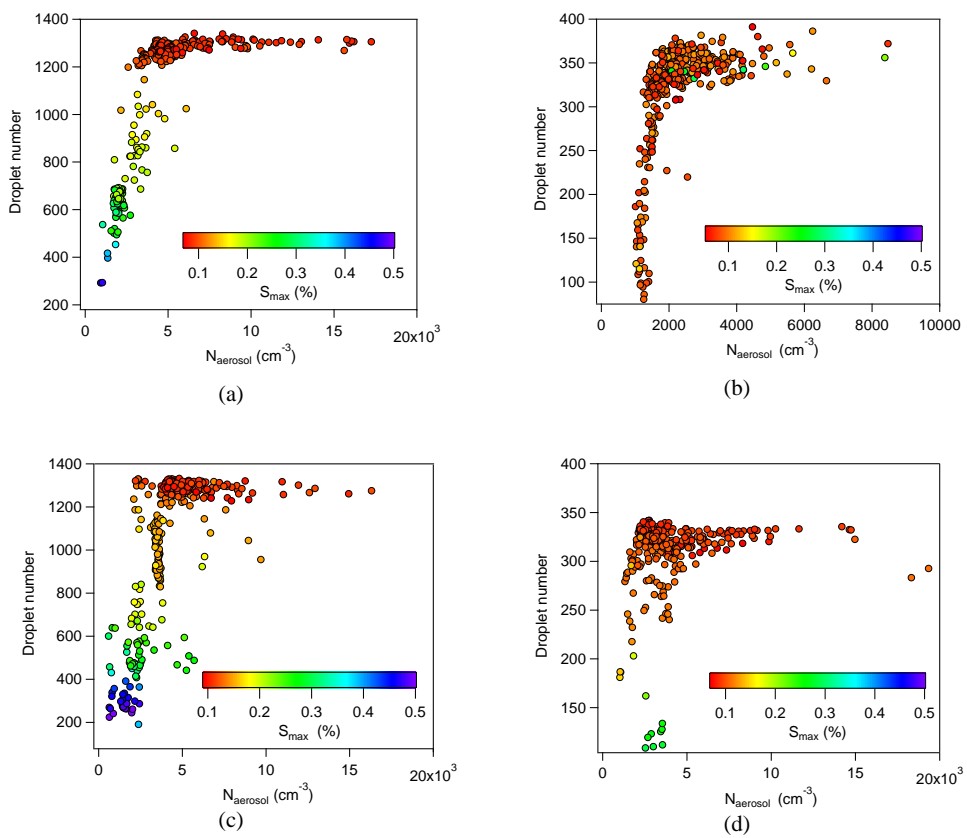

**Figure 4:** Cloud droplet number vs. total aerosol number for the derived characteristic vertical velocity (*w*\*). (a) for the Alabama sector during daytime (Flight 5) and (b) nighttime (Flight 15). (c) for Atlanta during daytime (Flight 6) and (d) nighttime (Flight 9). Data are colored by maximum supersaturation.





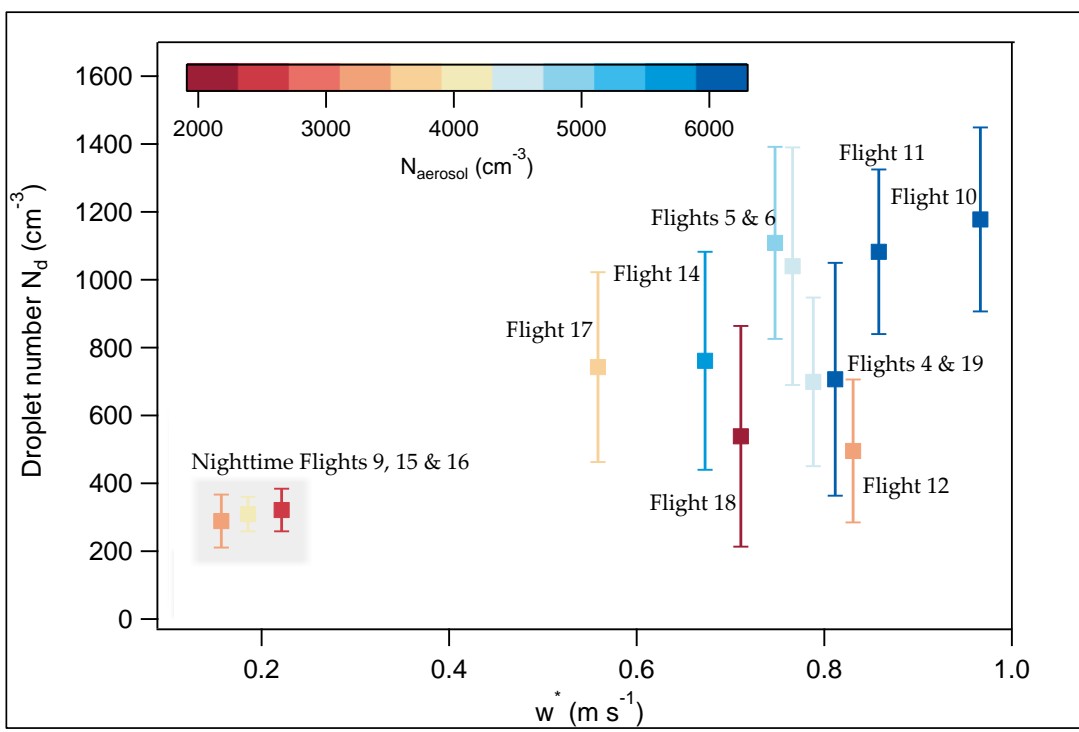

**Figure 5:** Average cloud droplet number vs. characteristic velocity during the 13 research flights, colored by total aerosol number. Error bars represent the standard deviation of cloud droplet number during each flight. The shaded area represents the flights conducted during nighttime.



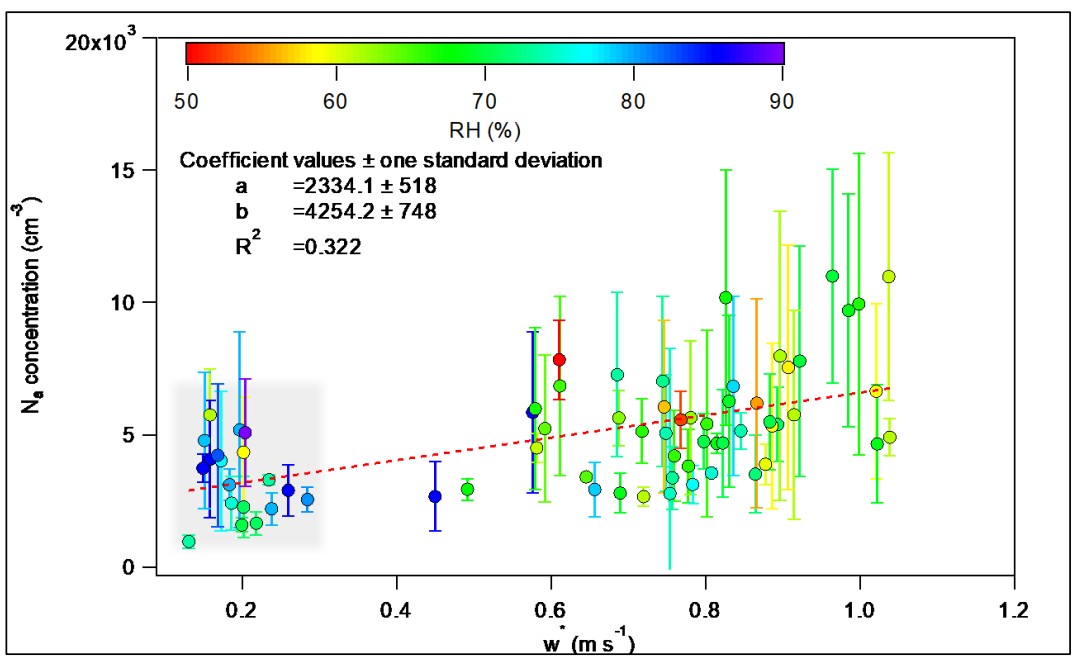

550

**Figure 6:** Total aerosol number vs. characteristic velocity during the segments of the flights when the
aircraft remained at a constant altitude within the boundary layer, colored by relative humidity. The shaded
area represents the segments of the flights conducted during nighttime.

554