# Peer review of "Drivers of cloud droplet number variability in the summertime 2 Southeast United States"

_Atmospheric Chemistry and Physics, 2020_

## Referee Comment (RC1) · Anonymous Referee #1 · 4 Apr 2020

Title: Drivers of cloud droplet number variability in the summertime Southeast United States Author(s): Aikaterini Bougiatioti, Athanasios Nenes, Jack J. Lin, Charles A. Brock, Joost de Gouw, Jin Liao, Ann M. Middlebrook, and Andre Welti MS No.: acp-2020-225

This manuscript focuses observations of aerosol and vertical velocities over the southeast US and how it impacts predicted cloud droplet number concentration. The work uses data from the 2013 field campaign SENEX with 13 flights over the Southeast US. They find that aerosol amount and vertical velocity are responsible for up to 90% of cloud droplet number variability. They stress, early in the manuscript, that most studies do not include the impact of vertical velocity. There are some edits required, though other than that it is a fell written manuscript that will be of interest for the aerosol com-

munity.

My recommendation to accept this work with revisions and modifications to figures.

Main comments: 1) Do you have access to actual cloud data? How do the calculated Nd values compare to the calculated Nd values presented in this paper? I find it hard to believe that there were no cloud data available. Even a simple discussion about how realistic the calculated Nd values are in comparison to what was seen in in situ observations is necessary.

2) For figures 5 and 6 there could be additional discussion in the manuscript. When looking at Figure 5: the first thought I had was it would be nice to see a comparison of cases when the sw* was the same and you could ese how droplet number and Na were related. That seems more important than looking at a range of Na and simultaneously looking at a range of w*. Maybe a three panel figure with "Low w*," "medium w*" and "high w*" like mentioned in the text but then plot Nd and Na? Secondly, for Figure 6, Could the difference in Na with w* be due to the vertical transport? Since the w* values are higher more aerosol can be brought up from the surface.

Table Comments: 1) All the tables are ok, though it might be helpful to note daytime vs. nighttime in some way, either by shading or some type of annotation (sun and moon perhaps?)

2) For Table 4: Perhaps add a mean row at the bottom for the contributions for K, Nd and sigmaw? A quick average gives k = 4.2, Nd = 75.2 and sigmaw = 13. Nd + sigmaw = 88.2. Is this where the 90% comes from that is mentioned in lines 313-314?

Figure Comments: 1) Figure 1: The 3D flight paths are hard to see in such a small format. Perhaps just 2D would be better, or make each panel larger. How many flights look like (a) and how many flights look like (b)? Could you include statistics about this? Otherwise it looks like your cherry picking examples.

2) Figure 2: Make all the panels larger, the legends are hard to read. Why are the
words and numbers together in the legends (e.g. Flight15 pass2)... spread them out Flight 15 pass 2. Also, in the caption Line 523: you say "flights" did you mean passes? In panel (c) you have Flight 14 pass 6 and in (b) you also have Flight 14 put pass 1. Also, on panel c) consider different colors for the lines. If someone was colorblind they would not be able to tell the difference between the pink/red lines and the greenish ones.

3) Figure 3: Suggestion: 4) Figure 4: same comment as in Figure 3: Add annotations to the figures to label the columns "Day" and "Night" and the rows "Alabama" and "Atlanta"

5) Figure 5: In the caption (line 547) "shading" is mentioned but is not visible in the figure. Also, the yellow marker for Flight 15 (I think) is difficult to see.

6) Figure 6: In the caption (line 552) "shading" is mentioned but is not visible in the figure. What is the "constant altitude" that is referred to in this figure? Include the altitude somehow.

Line by line comments: Line 37: Try not to use symbols in the abstract, just describe in words (it's clearer).

Line 45: remove "the" before "incoming"

Line 182: Specify Figure 1b here. Figure 1a does NOT show the significant decrease in organic mass fraction.

Line 236: how do you define what an "important contributor" is? What percentage do you consider important?

Line 242: Specify that the "first pair of flights" is for the Alabama flights.

Line 244: Specify that the "second pair of flights" is for the Atlanta flights.

Lines 253-256: The sentence that starts with "Figure 3" would be better up after "(see Fig 3.)" on line 240. It doesn't make sense where it is now.

Line 264: "characteristic", should be "characteristic,"

Line 313-314: How do you get the 90% number?

Line 319: "S.Atlantic" should be "Southeast Atlantic"

Line 523: you say "flights" did you mean passes? In panel (c) you have Flight 14 pass 6 and in (b) you also have Flight 14 put pass 1.

Line 527: add "calculated" between "showing" and "cloud"

---

## Referee Comment (RC2) · Anonymous Referee #2 · 22 Apr 2020

The authors use measurements of aerosol number and composition along with updraft variability to identify the role each plays in determining simulated cloud droplet number concentrations. I have many concerns with this manuscript. The authors reference anthropogenic and biogenic aerosol precursors as a possible driver of climate over the southeast united states, however there is little to no discussion of this feedback. Also, simulations of cloud droplet concentrations are not compared to any actual measurements of cloud droplet concentrations. One of the major key points of the manuscript relies on comparisons of night flights and day flights however there are only 3 night flights and a total of 10 day flights. It is hard to keep track of which cases are night time and which are day, though the diuranl variability of sigmaw is a key point of the paper. This makes it hard to follow this point. You refer to several different flights, and

honestly the flight number is somewhat meaningless to the reader. Referring to the flights by certain properties (I.e. night flight 1, night flight 2) would be more useful. The Figure quality is inconsistent and a few figures are repetitive, showing the same result in slightly different ways. Some Figures and tables do not contain data from all cases, leaving the reader to wonder why the other cases were omitted. It is not clear that any result came from section 3.1. Section 3.2 is confusing as it mainly involves a comparisons of individual cases and many sentences and paragraphs do not relate to one another. I was so lost that I stopped reading in this section. It is unclear what data (from tables) was used to calculate many of the numbers listed in this section.

The main result appears to be that updraft velocity and variability are higher during the day, leading to more "simulated" cloud droplets, which is not surprising or new. Comparisons to the contribution of organic mass and particle concentration is also not new. Overall, the manuscript lacks new and measurement supported results, lacks organization, contains figures of low quality, and hard to follow discussion. I am not suggesting rejects of the manuscript only because the measurements published are of high value. I suggest the manuscript be reconsidered after substantial revisions are made to the overall message and clarity of the text, and quality of the figures.

Specific comments:

Line 25: Different how? Explain how it is different before you talk about why.

Line 94: Can you provide a source for the WLOPC?

Line 194: I believe you meant to cite Table 1. What do you mean by "overall values"?

Line 196: For what? It would be helpful to lead the reader more currently it is hard to see where this text is going. Are these the distributions in Figure 2? If so cite them in this sentence.

198: I don't think you need this statement twice within 10 lines of eachother.

Line 203: You only chose 4 distributions for each plot in Figure 2. How did you choose

which flights to include/exclude? I suggest making your y axis the same for each flight. It would be more obvious that the concentrations are different. A log scale for the y axis may be helpful if the authors choose. At the very least please use the same notation for the y axis tick labels.

Line 205: " the modal diameters did not vary much" Why is that significant?

Line 209: You previously mentioned that the organic mass fraction was high during a night flight, but here you are saying 'contrasts between day and nighttime aerosol characteristics/variability may not be as large' Are you saying contrast in composition should be small between night and day? Are you saying the difference in accumulation mode concentration between night and day plays a bigger role in determining cloud droplet number concentration than aerosol a characteristics/variability? It is not clear and if you are saying the latter then you should reference you partial derivatives that you mentioned in line 164 to confirm. If you are going to "discuss the variability of the total aerosol number on droplet number in section 3.2" then it should probably not be mentioned here.

Line 212: It is not clear that "Cont kappa" and "Cont Na" is the partial derivative in Table 3/4. Be consistent with your abbreviations. "contribution" is listed as 'Cont' and 'Contrib' which is confusing.

Line 217: suggest changing "chemical composition" to kappa or hygroscopicitry parameter and if that is how "chemical composition" is expressed throughout the paper I suggest using one consistent term or symbol.

Line 228: reference table/figure that identifies daytime sigma2 varies little and is large. Sigma w at night seems to vary less than during the day based on your next two sentence.

Line 231-232: Is the data used to obtain 0.23+/-0.04 and 0.97+/-0.21in one of these tables?

Line 234: " total variability in Nd based on dNd/d$\kappa$, dNd/dNa and dNd/d$\sigma$w and the variances of $\kappa$, Na and $\sigma$w" this is repetitive.

Line 241: you should state these "sectors" were in atlanta and alabama respectively. You haven't referred to sectors at all so far, making it confusing to suddenly mention them. This paragraph is hard to follow. There are several numbers compared for different cases at different time periods

Line 257: these exact flights and "sectors" were discussed 2 paragraphs ago. This could be better organized.

Table 2: are times in local time? Why are some flights missing from this table? Is there a reason for the order in which flights are placed in the table?(flight 12 is listed after flight 14?)

Figure 3: your plot sizes are inconsistent. What are the hourglass markers? You should mention these are simulated droplet numbers.

Figure 4: Add units to the y axis label

---

## Author Response (AR1)

Atmos. Chem. Phys. Discuss., https://doi.org/10.5194/acp-2020-225 © Author(s) 2020. This work is distributed under the Creative Commons Attribution 4.0 License.

**Response to Anonymous Referee #1 comments**

This manuscript focuses observations of aerosol and vertical velocities over the southeast US and how it impacts predicted cloud droplet number concentration. The work uses data from the 2013 field campaign SENEX with 13 flights over the Southeast US. They find that aerosol amount and vertical velocity are responsible for up to 90% of cloud droplet number variability. They stress, early in the manuscript, that most studies do not include the impact of vertical velocity. There are some edits required, though other than that it is a fell written manuscript that will be of interest for the aerosol community.

My recommendation to accept this work with revisions and modifications to figures.

**Response:** We thank the anonymous referee for the thoughtful review. Suggestions and comments for the modification of the tables and figures addressed in the revised manuscript.

Main comments: 1) Do you have access to actual cloud data? How do the calculated  $N_d$  values compare to the calculated  $N_d$  values presented in this paper? I find it hard to believe that there were no cloud data available. Even a simple discussion about how realistic the calculated  $N_d$  values are in comparison to what was seen in in situ observations is necessary.

**Response:** Unfortunately, cloud data is not available as cloud sampling was avoided (the aircraft navigating in visual flight mode most of the time). It has been shown elsewhere (e.g. Kacarab et al., 2020 and others) that our droplet number calculation methodology gives good closure with observed droplet number.

2) For figures 5 and 6 there could be additional discussion in the manuscript. When looking at Figure 5: the first thought I had was it would be nice to see a comparison of cases when the sw\* was the same and you could see how droplet number and Na were related. That seems more important than looking at a range of Na and simultaneously looking at a range of w\*. Maybe a three panel figure with "Low w\*," "medium w\*" and "high w\*" like mentioned in the text but then plot Nd and Na? Secondly, for Figure 6, Could the difference in Na with w\* be due to the vertical transport? Since the w\* values are higher more aerosol can be brought up from the surface. *Response: These are great points. We have followed up on the reviewer's suggestion and splitting Figure 5 in three different graphs for low, medium and high w\* values, the covariance of the total aerosol number with the vertical velocity becomes even more apparent; for low w\* (during nighttime) changes in total aerosol number do not have a direct impact on calculated droplet number. On the other hand, for higher w\* there is a direct correlation between total aerosol number and droplet number, which for the highest observed w\* is even more accentuated, denoting*

the fact that the covariance of Na with w\* results in a higher variance in droplet number. Indeed, differences in Na with w\* can be partially due to the entrainment of more aerosol from the surface due to higher w\*, and this has been added in the revised manuscript. The respective discussion has been updated in the revised version.

In addition, we have included a discussion on the "limiting" droplet number that develops under high aerosol number, and its dependence on  $\sigma_w$  (shown in Figure 6). The implications of these findings are also discussed and quite interesting.

Table Comments: 1) All the tables are ok, though it might be helpful to note daytime vs. nighttime in some way, either by shading or some type of annotation (sun and moon perhaps?) *Response: Good suggestion! A sun and moon symbol is now added next to the number of each flight to denote whether it was a daytime or nighttime one.*

2) For Table 4: Perhaps add a mean row at the bottom for the contributions for K, Nd and sigmaw? A quick average gives k = 4.2, Nd = 75.2 and sigmaw = 13. Nd + sigmaw = 88.2. Is this where the 90% comes from that is mentioned in lines 313-314?

**Response:** We added a mean row with the respective averages for dNd/Nd and the contribution of each one of kappa, chemical composition and vertical velocity to the droplet number.

Figure Comments: 1) Figure 1: The 3D flight paths are hard to see in such a small format. Perhaps just 2D would be better, or make each panel larger. How many flights look like (a) and how many flights look like (b)? Could you include statistics about this? Otherwise it looks like your cherry picking examples.

**Response:** We replaced the 3D flight paths with 2D ones, showing the values of the organics mass fraction at the different altitudes throughout the flight. All figures will be added as supplementary material and a discussion about the similarity (or not) between flights is now added in the revised manuscript. We also fixed one of the color scales that was accidentally in reverse.

2) Figure 2: Make all the panels larger, the legends are hard to read. Why are the words and numbers together in the legends (e.g. Flight15 pass2): : : spread them out Flight 15 pass 2. Also, in the caption Line 523: you say "flights" did you mean passes? In panel (c) you have Flight 14 pass 6 and in (b) you also have Flight 14 put pass 1. Also, on panel c) consider different colors for the lines. If someone was colorblind they would not be able to tell the difference between the pink/red lines and the greenish ones.

**Response:** All panels and legends are now larger and words spread within each legend. Indeed in Figure 2 we have different passes of the same flight shown in different panels; in panel (c) we have Flight 14 pass 6; in (b) Flight 14 pass 1; in panel (b) Flight 10 pass 7; in panel (c) Flight 10 pass 3; in panel (a) Flight 11 pass 6; in panel (c) Flight 11 pass 1. The transects were often made at different altitudes, thus exhibiting different characteristics each time, which were subsequently compared to other similar passes. Different colors are now used for the lines in order to make them stand out more.

3) Figure 3: Suggestion: 4) Figure 4: same comment as in Figure 3: Add annotations to the figures to label the columns "Day" and "Night" and the rows "Alabama" and "Atlanta"

**Response:** We would like to thank the reviewer for the suggestion, the specific annotation for the columns and the rows are now added to both figures (Figure 3 and 4).

5) Figure 5: In the caption (line 547) "shading" is mentioned but is not visible in the figure. Also, the yellow marker for Flight 15 (I think) is difficult to see.

**Response:** The tinted background denoting nighttime flights is now darker, thus the marker for Flight 15 is now more easily visible.

6) Figure 6: In the caption (line 552) "shading" is mentioned but is not visible in the figure. What is the "constant altitude" that is referred to in this figure? Include the altitude somehow.

**Response:** The tinted background denoting nighttime flights is now darker. As far as the constant altitude is concerned, is it clear from Table 2 that it is not the same even within each flight, let alone between flights. We do not see how it would be easy to include this information in the graph.

Line by line comments: Line 37: Try not to use symbols in the abstract, just describe in words (it's clearer).

**Response:** We have left few symbols in the abstract, because we believe it helps with conveying our message more concisely.

Line 45: remove "the" before "incoming" *Response: Amended*

Line 182: Specify Figure 1b here. Figure 1a does NOT show the significant decrease in organic mass fraction.

**Response:** We have changed how Figure 1 is presented along with the accompanying discussion in the text.

Line 236: how do you define what an "important contributor" is? What percentage do you consider important?

**Response:** Values for  $\sigma_w$  during daytime flights are in the range of 0.7-1.22 with standard deviations between 0.07 and 0.31, while during nighttime flights the range of  $\sigma_w$  is of 0.2-0.33 and standard deviations <0.04. Therefore during a whole day the variation in  $\sigma_w$  values is more than a factor of 3.

Line 242: Specify that the "first pair of flights" is for the Alabama flights.

**Response:** Amended as follows:

"The first pair of flights were conducted over a rural area under moderate aerosol number conditions..."

Line 244: Specify that the "second pair of flights" is for the Atlanta flights.

**Response:** Amended

"... while the second pair exhibited somewhat higher aerosol numbers owing to its proximity to the Atlanta metropolitan area."

Lines 253-256: The sentence that starts with "Figure 3" would be better up after "(see Fig 3.)" on line 240. It doesn't make sense where it is now.

**Response:** We would like to thank the reviewer for pointing out this inconsistency. The description of Fig. 3 is now moved to L259 where Fig. 3 was introduced.

Line 264: "characteristic", should be "characteristic,"

**Response:** Section 3.2 has been rewritten, and the sentence is question no longer appears in the revised text.

Line 313-314: How do you get the 90% number?

**Response:** When adding the contribution of  $N_a$  and  $\sigma_w$  to the variability of the total droplet number, for each flight this added contribution is more than 90%.

Line 319: "S.Atlantic" should be "Southeast Atlantic" *Response: Amended*

Line 523: you say "flights" did you mean passes? In panel (c) you have Flight 14 pass

6 and in (b) you also have Flight 14 put pass 1. *Response: Indeed so, amended*

Line 527: add "calculated" between "showing" and "cloud" *Response: Amended*

Atmos. Chem. Phys. Discuss., https://doi.org/10.5194/acp-2020-225 © Author(s) 2020. This work is distributed under the Creative Commons Attribution 4.0 License.

**Response to Anonymous Referee #2 comments**

The authors use measurements of aerosol number and composition along with updraft variability to identify the role each plays in determining simulated cloud droplet number concentrations. I have many concerns with this manuscript. The authors reference anthropogenic and biogenic aerosol precursors as a possible driver of climate over the southeast united states, however there is little to no discussion of this feedback. Also, simulations of cloud droplet concentrations are not compared to any actual measurements of cloud droplet concentrations.

**Response**: We thank the referee for the thoughtful and careful review. Some of the issues were also raised by Reviewer #1 and are addressed hereafter. The reference to anthropogenic and biogenic precursors was to provide the context for the SENEX campaign. The emissions-aerosol-cloud link, although very important, is not the focus of this study. Here we focus primarily on the aerosol-cloud link, and the underappreciated role of vertical velocity covariance. Unfortunately, there were no cloud data available, as the aircraft was operating in visual flight mode most of the time. However, we have shown in other studies (e.g. Kacarab et al., 2020) that our approach for calculating Nd values agree with observed droplet number in non-precipitating boundary layer clouds, therefore the conclusions are robust.

One of the major key points of the manuscript relies on comparisons of night flights and day flights however there are only 3 night flights and a total of 10 day flights. It is hard to keep track of which cases are night time and which are day, though the diuranl variability of sigmaw is a key point of the paper. This makes it hard to follow this point. You refer to several different flights, and honestly the flight number is somewhat meaningless to the reader. Referring to the flights by certain properties (I.e. night flight 1, night flight 2) would be more useful.

**Response**: This issue was also raised by Reviewer #1. We now more clearly distinguish between daytime and nighttime flights (sun for day and moon for night) so the diurnal variability of  $\sigma w$  is easier to see. Finally, the added characterization of the two pairs of flights as "Urban" and "Rural" now facilitates keeping track of which flights are mentioned.

The Figure quality is inconsistent and a few figures are repetitive, showing the same result in slightly different ways. Some Figures and tables do not contain data from all cases, leaving the reader to wonder why the other cases were omitted.

**Response**: This was pointed out by Reviewer #1 as well, and now Supplementary material includes organics mass fractions and estimated cloud droplet number for all studied flights. Furthermore, new figures are now plotted in the revised version which point out important findings for the vertical velocity vs. aerosol limiting regimes and figures from the previous version showing similar results are now moved to the supplementary material.

It is not clear that any result came from section 3.1. Section 3.2 is confusing as it mainly involves a comparisons of individual cases and many sentences and paragraphs do not relate to one another. I was so lost that I stopped reading in this section. It is unclear what data (from tables) was used to calculate many of the numbers listed in this section.

**Response**: These sections have been rewritten for clarity (with additional analysis) and numbersfigures are cited in the supporting discussion.

The main result appears to be that updraft velocity and variability are higher during the day, leading to more "simulated" cloud droplets, which is not surprising or new. Comparisons to the contribution of organic mass and particle concentration is also not new. Overall, the manuscript lacks new and measurement supported results, lacks organization, contains figures of low quality, and hard to follow discussion. I am not suggesting rejects of the manuscript only because the measurements published are of high value. I suggest the manuscript be reconsidered after substantial revisions are made to the overall message and clarity of the text, and quality of the figures.

**Response:** Most aspects of warm cloud physics and especially droplet formation are known for decades. However, droplet formation remains at the heart of the aerosol indirect effect, so ensuring that models capture droplet number for the "right reasons" is critical for constraining aerosol-cloud-climate interactions. The latter aspect is where a huge knowledge and data gap exists – and where our study provides important constraints (vertical velocity, aerosol number, potential droplet number) and insights (covariance between  $\sigma w$  and Na and their role on the Na-Nd relationship in the SE US). The additional insights on the limiting droplet number, and its explicit dependence on  $\sigma_w$  is also new and important, and offers a new possibility for remote sensing. Given that model assessments of aerosol–cloud-climate interactions do not evaluate for vertical velocity (or covariance with other parameters), our work here shows that this can lead to an unresolved source of hydrometeor variability and bias. We have made these points very clear now.

Specific comments:

Line 25: Different how? Explain how it is different before you talk about why. *Response: The abstract has been rewritten and no longer includes a reference to differing climate trends.*

Line 94: Can you provide a source for the WLOPC? *Response:* We have added Brock et al. (2011) as a source.

Line 194: I believe you meant to cite Table 1. What do you mean by "overall values"? **Response:** We thank the reviewer for pointing out this inconsistency. The overall value is the  $0.25\pm0.05$  stated just before, a row has been added in Table 1 giving the average values and a clarification is added to the revised text.

Line 196: For what? It would be helpful to lead the reader more currently it is hard to see where this text is going. Are these the distributions in Figure 2? If so cite them in this sentence.

**Response:** Aerosol number size distributions are crucial for the calculation of the total aerosol number during each flight, as they enter as input in the droplet number parameterization. This is added as a clarification in the revised text.

198: I don't think you need this statement twice within 10 lines of each other. *Response: The repetition in now omitted from the revised version.*

Line 203: You only chose 4 distributions for each plot in Figure 2. How did you choose which flights to include/exclude? I suggest making your y axis the same for each flight. It would be more obvious that the concentrations are different. A log scale for the y axis may be helpful if the authors choose. At the very least please use the same notation for the y axis tick labels.

**Response:** These are good points. Each plot represents a grouping based on e.g., passes in free tropospheric conditions (a) or nighttime flights (d). The selection of the different pass types reflects the need to represent daytime-nighttime contrasts (which is important for droplet number calculations as shown below) as well as the shape of the size distribution. As each flight had around 5 passes in a constant altitude (e.g., Table 2) presenting all average particle size distributions in one graph is cumbersome. Instead size distributions are grouped with common characteristics, and the data from 9/13 flights are represented in the figure, while the rest mostly fall in one of the represented categories. The vertical axes for the three plots are now similar (apart from the nighttime flights, which is maintained different because during nighttime the aerosol number as well as the variability was lower).

Line 205: " the modal diameters did not vary much" Why is that significant? *Response: This is important as it dictates particles of which mode will activate, depending on S*max.

Line 209: You previously mentioned that the organic mass fraction was high during a night flight, but here you are saying 'contrasts between day and nighttime aerosol characteristics/variability may not be as large' Are you saying contrast in composition should be small between night and day? Are you saying the difference in accumulation mode concentration between night and day plays a bigger role in determining cloud droplet number concentration than aerosol a characteristics/variability? It is not clear and if you are saying the latter then you should reference you partial derivatives that you mentioned in line 164 to confirm. If you are going to "discuss the variability of the total aerosol number on droplet number in section 3.2" then it should probably not be mentioned here.

**Response:** Large part of the discussion has been revised to promote a more coherent and comprehensive flow in the text.

Line 212: It is not clear that "Cont kappa" and "Cont Na" is the partial derivative in Table 3/4. Be consistent with your abbreviations. "contribution" is listed as 'Cont' and 'Contrib' which is confusing.

**Response:** We sought to determine the relative contribution of aerosol composition (expressed by  $\kappa$ ), total aerosol number and vertical velocity to variations of droplet number, using a variancebased approach. For this, we compute the partial sensitivities of droplet number to  $N_a$ ,  $\kappa$ ,  $\sigma_w$ (Sullivan et al., 2016; Bougiatioti et al., 2017), multiply them with their respective variance and sum as follows to obtain the droplet number variance:

$$\sigma^2 N_d = \left(\frac{\overline{\partial N_d}}{\partial N_a} \sigma N_a\right)^2 + \left(\frac{\overline{\partial N_d}}{\partial \kappa} \sigma \kappa\right)^2 + \left(\frac{\overline{\partial N_d}}{\partial \sigma_w} \sigma \sigma_w\right)^2$$

The relative contribution of  $N\alpha$ ,  $\kappa$ , and  $\sigma w$  to the droplet number variance is then estimated as follows, and their values presented in Tables 3 and 4:

$$\varepsilon_{Na} = \frac{\left(\frac{\overline{\partial N_d}}{\partial N_a} \sigma N_a\right)^2}{\sigma^2 N_d}, \varepsilon_{\kappa} = \frac{\left(\frac{\overline{\partial N_d}}{\partial \kappa} \sigma \kappa\right)^2}{\sigma^2 N_d}, \varepsilon_{\sigma w} = \frac{\left(\frac{\overline{\partial N_d}}{\partial \sigma_w} \sigma \sigma_w\right)^2}{\sigma^2 N_d}$$

This is further clarified in the revised text and all abbreviations are now consistent.

Line 217: suggest changing "chemical composition" to kappa or hygroscopicity parameter and if that is how "chemical composition" is expressed throughout the paper I suggest using one consistent term or symbol.

**Response:** Changes in the hygroscopicity parameter are a direct result of chemical composition changes, and we stress this point (e.g. L230 changes in hygroscopicity (i.e. chemical composition)). Keeping the composition-hygroscopicity link is important, given that variations in hygroscopicity (chemical composition) induce variability in droplet number.

Line 228: reference table/figure that identifies daytime sigma2 varies little and is large. Sigma w at night seems to vary less than during the day based on your next two sentence.

**Response:** This is now clarified in the text.

"The large diurnal variability in  $\sigma_w$  (from 0.3 m s-1 at night to 1.0 m s-1 at day) contributes considerably to the diurnal variability in  $N_{d}$ ...."

Line 231-232: Is the data used to obtain 0.23±0.04 and 0.97±0.21 in one of these tables? *Response: This is now clarified in the text.*

"The vertical velocity distributions observed gave  $\sigma_w = 0.97 \pm 0.21 \text{ m s}^{-1}$  for daytime flights, and  $\sigma_w = 0.23 \pm 0.04 \text{ m s}^{-1}$  for nighttime flights (Table 2 and SP3)."

Line 234: " total variability in Nd based on dNd/d\_, dNd/dNa and dNd/d\_w and the variances of \_, Na and \_w" this is repetitive.

**Response:** Amended

"...we estimate their contribution to the total variability in  $N_d$  based on the variances of  $\kappa$ ,  $N_a$  and  $\sigma_w$  and the sensitivity of droplet formation to those parameters."

Line 241: you should state these "sectors" were in atlanta and alabama respectively. You haven't referred to sectors at all so far, making it confusing to suddenly mention them. This paragraph is hard to follow. There are several numbers compared for different cases at different time periods *Response:* Good point. They different areas are now stated. The paragraph has now been rewritten.

Line 257: these exact flights and "sectors" were discussed 2 paragraphs ago. This could be better organized.

**Response:** Section 3.2 has been rewritten, in response to this (and other similar) comments.

Table 2: are times in local time? Why are some flights missing from this table? Is there a reason for the order in which flights are placed in the table? (flight 12 is listed after flight 14?)

**Response:** The table header now clarifies that we refer to local time. The table contains the most relevant data from the flights that are used in the text. All flights with all segments and relevant characteristics ( $\sigma_w$ ,  $w^*$  and altitude) are available in the supplementary file accompanying the manuscript. The reason flight 12 is listed after 14 is simply for aesthetic reasons.

Figure 3: your plot sizes are inconsistent. What are the hourglass markers? You should mention these are simulated droplet numbers.

**Response:** Thank you for pointing this out, all changes made. Additional information is also included in the supplement.

Figure 4: Add units to the y axis label *Response: Amended*.

**Drivers of cloud droplet number variability in the summertime Southeast United States**

Aikaterini Bougiatioti1,2, Athanasios Nenes2,3,4, Jack J. Lin2,a, Charles A. Brock5, Joost A. de Gouw5,6,b, Jin
 Liao5,6,c,d, Ann M. Middlebrook5, André Welti5,6,e

[revised manuscript text omitted]
 proportional change in  $N_d$  (factor of 3.6 and at). These responses however occur 389 over the same timediurnal timescale, during which  $N_a$  also changes; the  $N_{d}$ -covariance between  $\sigma_w$  with  $N_a$ 390 391 enhances the apparent response of  $N_d$  to different changes in  $N_a$  levels may be enhanced by a factor of 5 (Figure 4).  $S_{max}$  changes in response to aerosol concentration, in a way that tends to partially mitigate  $N_d$ 392 393 responses to aerosol. Overall, maximum supersaturation levels remain quite low (0.14±0.05%) with 394 predicted levels being much lower in lower altitudes (0.05±0.1%). Because of the strong competition for water vapor (expressed by the low Smar), cloud droplet number exhibits enhanced sensitivity to aerosol 395 396 number variations throughout the flights, regardless of aerosol composition. On the other hand, droplet 397 concentration especially within the boundary layer approaches a "plateau" that is strongly driven by vertical 398 velocity (turbulence) and the resulting supersaturation, but also aerosol concentration. In "cleaner" 399 environments where total aerosol number is lowernot impacted by local sources, the relative 400 contribution response of vertical velocity  $N_d$  to cloud droplet number  $\sigma_w$  is almost half during nighttime (24%) vs. 42% twice as great at night than during the day (24% for daytime) while Flight 5 vs. 42% for nighttime 401 402 Flight 15). On the other hand, the relative contribution of  $N_a$  to the variance in response of  $N_d$  to  $N_a$  is somewhat higher (69% vs. 51% slightly lower during daytime) even though Na is 2-fold lower the night than 403 during night. On the contrary, in 
[revised manuscript text omitted]

- 615 fundamental understanding of the Rolerole of Aerosol-Cloud Interactionsaerosol-cloud interactions in
- 616 the Climate System, Pclimate system, Proc. Nat. Acad. Sci. USA, 113, 5781–5790,
   617 https://doi.org/10.1073/pnas.1514043113, 2016.
- Sullivan, S.C., Lee, D., Oreopoulos, L., and Nenes, A.: The role of updraft velocity in temporal variability
  of cloud hydrometeor number, Proc. Nat. Acad. Sci, 113, 21, https://doi-.org/10.1073/pnas.1514039113, 2016.
- Wagner, N. L., Brock, C. A., Angevine, W. M., Beyersdorf, A., Campuzano-Jost, P., Day, D., de Gouw, J.
  A., Diskin, G. S., Gordon, T. D., Graus, M. G., Holloway, J. S., Huey, G., Jimenez, J. L., Lack, D. A.,
- 623 Liao, J., Liu, X., Markovic, M. Z., Middlebrook, A. M., Mikoviny, T., Peischl, J., Perring, A. E.,
- 624 Richardson, M. S., Ryerson, T. B., Schwarz, J. P., Warneke, C., Welti, A., Wisthaler, A., Ziemba, L.
- D., and Murphy, D. M.: In situ vertical profiles of aerosol extinction, mass, and composition over the

- southeast United States during SENEX and SEAC4RS: observations of a modest aerosol enhancement
  aloft, Atmos. Chem. Phys., 15, 7085-7102, https://doi.org/10.5194/acp-15-7085-2015, 2015.
- 628 Warneke, C., Trainer, M., de Gouw, J. A., Parrish, D. D., Fahey, D. W., Ravishankara, A. R., Middlebrook,
- A. M., Brock, C. A., Roberts, J. M., Brown, S. S., Neuman, J. A., Lerner, B. M., Lack, D., Law, D.,

630 Hübler, G., Pollack, I., Sjostedt, S., Ryerson, T. B., Gilman, J. B., Liao, J., Holloway, J., Peischl, J.,

- 631 Nowak, J. B., Aikin, K. C., Min, K.-E., Washenfelder, R. A., Graus, M. G., Richardson, M., Markovic,
- 632 M. Z., Wagner, N. L., Welti, A., Veres, P. R., Edwards, P., Schwarz, J. P., Gordon, T., Dube, W. P.,
- 633 McKeen, S. A., Brioude, J., Ahmadov, R., Bougiatioti, A., Lin, J. J., Nenes, A., Wolfe, G. M., Hanisco,
- 634 T. F., Lee, B. H., Lopez-Hilfiker, F. D., Thornton, J. A., Keutsch, F. N., Kaiser, J., Mao, J., and Hatch,
- 635 C. D.: Instrumentation and measurement strategy for the NOAA SENEX aircraft campaign as part of
- 636 the Southeast Atmosphere Study 2013, Atmos. Meas. Tech., 9, 3063-3093,
  637 https://doi.org/10.5194/amt-9-3063-2016, 2016.
- Wilson, J. C., Lafleur, B. G., Hilbert, H., Seebaugh, W. R., Fox, J., Gesler, D. W., Brock, C. A., Huebert, 638 B. J., and Mullen, J.: Function and performance of a low turbulence inlet for sampling supermicron 639 640 particles from aircraft platforms, Aerosol Sci. Tech., 38, 790-802, 641 https://doi.org/10.1080/027868290500841, 2004.
- Weber RJ, Guo H, Russell AG, Nenes A.: High aerosol acidity despite declining atmospheric sulfate
  concentrations over the past 15 years, Nat. Geosci., 9, 282-285,
  2016.https://doi.org/10.1038/ngeo2665, 2016.
- 645 Yu, S., Alapaty, K., Mathur, R., Pleim, J., Zhang, Y., Nolte, C., Eder, B., Foley, K., and Nagashima, T.:
- Attribution of the United States "warming hole": Aerosol indirect effect and precipitable water vapor.
- 647 Sci. Rep., 4, 6929, https://doi.org/10.1038/srep06929, 2014.

Table 1: Research flights from the SENEX 2013 campaign used in this study. The symbol "\$\$\vec{\$\phi\$}" next to
 each flight number refers to daytime flight, and "€" refers to a nighttime flight.

| Flight      | Date | Local Time              | #Hygroscopicity | Organic mass     |
|-------------|------|-------------------------|-------------------------------|------------------|
|             |      | ( <del>CDT,</del> UTC-5 | Parameter K     | fraction         |
|             |      | hrs)                    |                               |                  |
| 4\$         | 10/6 | 09:55-16:30             | 0.23±0.02                     | $0.62 \pm 0.11$  |
| 5¢          | 11/6 | 11:30-17:57             | 0.20±0.00                     | $0.68{\pm}0.05$  |
| 6\$         | 12/6 | 09:48-15:31             | 0.21±0.01                     | $0.68{\pm}0.07$  |
| 9 C  | 19/6 | 17:30-23:29             | 0.24±0.01                     | $0.66 \pm 0.06$  |
| 10\$        | 22/6 | 10:01-17:09             | 0.21±0.02                     | $0.68{\pm}0.08$  |
| 11\$        | 23/6 | 10:08-17:22             | 0.25±0.03                     | $0.58{\pm}0.07$  |
| 12\$        | 25/6 | 10:18-17:25             | 0.39±0.02                     | 0.35±0.18        |
| 14\$        | 29/6 | 10:26-17:39             | 0.22±0.03                     | $0.62{\pm}0.07$  |
| 15€         | 2/7  | 20:08-02:51             | 0.28±0.05                     | 0.55±0.09        |
| 16 C | 3/7  | 19:56-02:55             | 0.22±0.05                     | 0.67±0.09        |
| 17\$        | 5/7  | 09:52-16:24             | 0.23±0.05                     | 0.59±0.14        |
| 18\$        | 6/7  | 09:19-16:18             | 0.31±0.02                     | $0.52{\pm}0.08$  |
| 19\$        | 8/7  | 10:11-16:44             | 0.23±0.04                     | $0.62{\pm}0.08$  |
| Average     |      |                         | 0.25±0.05              | 0.60±0.09 |

**Table 2:** Flight number, time interval, spectral dispersionstandard deviation of vertical wind velocity ( $\sigma_w$ ) and characteristic vertical velocity  $w^*=0.79\sigma_w$  during flight segments where the aircraft flew at a constant altitude.

|   | Flight
(pass) | Time
<del>Range</del> Interval |                                 |                            | Altitude
a.s.l. (m) | Flight
(pass) | Time
<del>Range</del> |                                 |                                | Altitude a.s.l.
(m) |
|---|------------------|-----------------------------------|---------------------------------|----------------------------|------------------------|------------------|--------------------------|---------------------------------|--------------------------------|-------------------------------|
|   | u /              | (Local Time)               | $\sigma_w$ (m s -1 ) | w*
(m s -1 ) |                        |                  | Interval
(Local Time) | $\sigma_w$ (m s -1 ) | w*
(m s -1 )     |                               |
|   | 5(1)             | 12:31-12:58                       |                                 |                            | $549\pm58$             | 9(1)             | 18:44-18:58              | 0.255                           | 0. <del>202</del>              | 797±2.01                      |
|   |                  |                                   | 1.02                            | 0.81                       |                        |                  |                          | 25                       | 20                      |                               |
|   | 5 (2)            | 13:16-13:29                       | 0.82                            | 0.65                       | 982±11                 | 9 (2)            | 19:20-19:29              | 0. <del>249</del>               | 0. <del>197</del>              | 740±1.23                      |
| - | 5 (3)            | 13.34-13.50                       | 0.82                            | 0.03                       | 502+13                 | 9 (3)            | 10.33_10.48              | 0.217                           | $\frac{4}{0.171}$              | 740+1.23                      |
|   | 5 (5)            | 15.54-15.50                       | 1.01                            | 0.80                       | 502±15                 | 9(3)             | 17.33-17.40              | 22                       | 17                      | /40±1.23                      |
|   | 5 (4)            | 13:53-14:08                       |                                 |                            | 614±27                 | 9 (4)            | 19:51-20:25              | 0. <del>218</del>               | 0.173                          | 776±1.22                      |
| - | - (-)            |                                   | 1.03                            | 0.81                       | (0.0.10                |                  |                          | 22                       | 17                      |                               |
|   | 5 (5)            | 14:20-15:00                       | 0.91                            | 0.72                       | 603±40                 | 9 (5)            | 20:34-20:39              | $0.\frac{232}{23}$              | $0.\frac{183}{18}$             | 597±1.19                      |
|   | 5 (6)            | 15:35-15:41                       |                                 |                            | 533±18                 | 9 (7)            | 20:56-21:10              | 0. <del>201</del>               | 0. <del>158</del>              | 773±1.11                      |
| _ |                  |                                   | 0.87                            | 0.69                       |                        |                  |                          | 20                       | 16                      |                               |
|   | 5 (7)            | 16:17-16:30                       | 0.77                            | 0.61                       | 638±23                 | 9 (8)            | 21:31-21:45              | 0. <del>191</del>
19  | 0. <del>151</del>
15 | 725±1.18                      |
|   | 5 (8)            | 16:31-16:39                       |                                 |                            | 559±18                 | 9 (9)            | 22:24-22:31              | 0.257                           | 0. <del>203</del>              | 745±1.36                      |
| - | 5 (0)     | 12.10.12.00                       | 0.55                            | 0.44                       | (0() 10                | 0 (10)           | 22 40 22 54              | 26                       | 20                      | 004:107                       |
|   | 5 (9)            | 17:10-17:22                       | 0.53                            | 0.42                       | 686±40                 | 9 (10)           | 22:48-22:54              | 0.221
22                     | 0. <del>175</del>
17 | 804± 1.37                     |
|   | 14(1)            |                                   |                                 |                            | 558±2                  | 15(1)            | 21:09-21:52              | 0. <del>236</del>               | 0. <del>186</del>              | 505±6.64                      |
| - |                  | 12:34-12:49                       | 0.94                            | 0.75                       |                        |                  |                          | 24                       | 19                      |                               |
|   | 14 (2)           | 13:57-14:17                       | 0.97                            | 0.77                       | 658±3                  | 15 (2)           | 22:19-22:31              | 0. <del>301</del>
30         | 0. <del>238</del>
24        | 633±1.21                      |
| - | 14 (3)           |                                   | ,                               |                            | 737±3                  | 15 (3)           | 22:42-22:54              | 0.255                           | 0.202                          | 600±1.17                      |
|   |                  | 14:22-14:46                       | 0.95                            | 0.75                       |                        |                  |                          | 25                       | 20                      |                               |
|   | 14 (4)           | 14 50 15 22                       | 0.55                            | 0.42                       | 746±23                 | 15 (4)           | 23:26-23:37              | 0. <del>329</del>               | 0. <del>260</del>              | 908±1.56                      |
| - | 14 (5)           | 14:58-15:33                       | 0.55                            | 0.43                       | 714+2                  | 15 (5)           | 00.02 00.10              | 33                       | 26                      | 1208+1.22                     |
|   | 14 (3)           | 15:55-16:08                       | 0.57                            | 0.45                       | /14±3                  | 15(5)            | 00:02-00:19              | $\frac{0.297}{30}$              | $\frac{0.233}{23}$             | 1208±1.23                     |
|   | 14 (6)           |                                   |                                 |                            | 801±3                  | 15 (6)           | 00:43-1:08               | 0.253                           | 0. <del>199</del>              | 592±1.37                      |
| - |                  | 16:11-16:21                       | 0.77                            | 0.61                       |                        |                  |                          | 25                       | 20                      | (= ( ) ) )                    |
|   | 14 (7)           | 16:33-16:41                       | 0.45                            | 0.35                       | 793±2                  | 15 (7)           | 1:10-1:24                | $0.\frac{276}{28}$              | 0.218
22                    | 676±1.02                      |
|   |                  |                                   |                                 |                            |                        | 15 (8)           | 1:37-2:02                | 0. <del>207</del>               | 0. <del>164</del>              | 713±19.5                      |
|   |                  |                                   |                                 |                            |                        |                  |                          | 21                       | 16                      |                               |
|   | 12 (1)           | 11:50-12:34                       | 0.96                            | 0.75                       | 484±3                  | 19(1)            | 11:20-11:41              | 0. <del>622</del>
62         | 0.4 <del>92</del>
49        | 1014±2.27                     |
|   | 12 (2)           |                                   |                                 |                            | 503±3                  | 19 (2)           | 12:09-12:23              | 1. <del>203</del>               |                                | 652±3.34                      |
|   | ~ /              | 12:48-13:18                       | 1.09                            | 0.86                       |                        |                  |                          | 20                       | 0.95                           |                               |
|   | 12 (3)           | 10.04.10.50                       | 1.10                            | 0.00                       | 894±3                  | 19 (3)           | 12:51-13:10              | 0. <del>873</del>               | 0. <del>689</del>              | 537±2.51                      |
|   |                  | 13:34-13:50                       | 1.12                            | 0.88                       |                        |                  |                          | 87                              | 69                      |                               |

| 12 (4) |             |      |      | 479±4 | 19 (4) | 13:22-13:49 | 1. <del>294</del> | 1. <del>022</del> | 518±22.6 |
|--------|-------------|------|------|-------|--------|-------------|-------------------|-------------------|----------|
|        | 14:06-14:40 | 1.04 | 0.82 |       |        |             | 29         | 02         |          |
| 12 (5) |             |      |      | 521±3 | 19 (5) | 14:44-14:57 | 1. <del>361</del> | 1. <del>075</del> | 528±3.26 |
|        | 15:21-15:32 | 1.10 | 0.87 |       |        |             | 36         | 07         |          |
| 12 (6) |             |      |      | 475±3 | 19 (6) | 15:04-16:06 | 0. <del>896</del> | 0. <del>708</del> | 524±2.8  |
|        | 15:43-16:02 | 0.99 | 0.78 |       |        |             | 90         | 71         |          |

**Table 3:** Derived cloud parameters (maximum supersaturation, droplet number) and relative contribution of chemical composition and total

aerosol number for different vertical velocities. Numbers in parentheses indicate standard deviation values. The symbol "\$" next to each flight number refers to daytime flight, and "(" refers to a nighttime flight.

661

| Flight | Na   | Std        |        | $\sigma_w=0.1 \text{ m}$ | s -1 |         | $\sigma_{w}=0.3 \text{ m s}^{-1}$ |              |         | $\sigma_{\rm W}=0.6 {\rm ~m~s^{-1}}$ |        |              |         | $\sigma_w = 1.0 \text{ m s}^{-1}$ |        |       |         |         |
|--------|------|-------------------|--------|--------------------------|-----------------|---------|-----------------------------------|--------------|---------|--------------------------------------|--------|--------------|---------|-----------------------------------|--------|-------|---------|---------|
| 0      |      | Day N             | Smax   | Nd                       | Contrib         | Contrib | Smax                              | Nd           | Contrib | Contrib                              | Smax   | Nd           | Contrib | Contrib                           | Smax   | Nd    | Contrib | Contrib |
|        |      | Dev Iva    |        |                          | κ               | Na      |                                   |              | κ       | Na                                   |        |              | κ       | Na                                |        |       | κ       | Na      |
|        |      | <del>variab</del> |        |                          |                 |         |                                   |              |         |                                      |        |              |         |                                   |        |       |         |         |
| 40     | 6118 | 4520              | 0.11   | 122                      | 0.08            | 0.92    | 0.16                              | 315          | 0.20    | 0.80                                 | 0.21   | 520          | 0.23    | 0.77                              | 0.26   | 737   | 0.2     | 0.8     |
|        |      |                   | (0.06) | (41)                     |                 |         | (0.09)                            | (114)        |         |                                      | (0.12) | (212)        |         |                                   | (0.17) | (321) |         |         |
| 5₽     | 4324 | 2598              | 0.08   | 139                      | 0.09            | 0.91    | 0.1                               | 388          | 0.15    | 0.85                                 | 0.14   | 712          | 0.17    | 0.8308                     | 0.17   | 1063  | 0.21    | 0.79    |
|        |      |                   | (0.04) | (31)                     |                 |         | (0.06)                            | (104)        |         |                                      | (0.08) | (216)        |         | 3                          | (0.1)  | (360) |         |         |
| 6\$    | 4958 | 3054              | 0.07   | 151                      | 0.03            | 0.97    | 0.08                              | 422          | 0.11    | 0.89                                 | 0.1    | 773          | 0.08    | 0.92                              | 0.13   | 1162  | 0.07    | 0.93    |
|        |      |                   | (0.07) | (24)                     |                 |         | (0.04)                            | (70)         |         |                                      | (0.06) | (171)        |         |                                   | (0.07) | (302) |         |         |
| 9⊄     | 4271 | 3095              | 0.07   | 152                      | 0.05            | 0.95    | 0.12                              | 367          | 0.17    | 0.83                                 | 0.16   | 533          | 0.17    | 0.83                              | 0.19   | 680   | 0.12    | 0.88    |
|        |      |                   | (0.02) | (18)                     |                 |         | (0.04)                            | (68)         |         |                                      | (0.05) | (115)        |         |                                   | (0.06) | (126) |         |         |
| 10\$   | 6286 | 7201              | 0.07   | 158                      | 0.02            | 0.98    | 0.1                               | 422          | 0.02    | 0.98                                 | 0.14   | 748          | 0.04    | 0.96                              | 0.18   | 1063  | 0.09    | 0.91    |
|        |      |                   | (0.03) | (24)                     |                 |         | (0.05)                            | (86)         |         |                                      | (0.07) | (180)        |         |                                   | (0.08) | (295) |         |         |
| 110    | 5969 | 7271              | 0.04   | 137                      | 0.01            | 0.99    | 0.06                              | 381          | 0.04    | 0.96                                 | 0.08   | 695          | 0.03    | 0.97                              | 0.10   | 1025  | 0.03    | 0.97    |
|        |      |                   | (0.01) | (19)                     |                 |         | (0.01)                            | (61)         |         |                                      | (0.02) | (134)        |         |                                   | (0.02) | (226) |         |         |
| 12\$   | 3154 |

---

## Author Response (AR2)

Reviewer comments:

The authors have sufficiently responded to reviewer comments, improving figures, improving the clarity, making it clear what cases are night/day and rural/urban, and clearly indicating the importance of these results.

*Response: We thank the anonymous referee for the thoughtful review. The proposed suggestions and comments for the modification of the manuscript helped improve the quality of the manuscript.*

The figures have been improved however, I highly recommend using images for your plots instead of vector graphics. Especially in the supplement where there are so many vector graphics. Currently it takes a while to load each figure and it is cumbersome to scroll through the document.

*Response: Indeed, all figures now in the supplement are in .jpeg format so that all figures are loaded rapidly and it doesn't take long to scroll through the document.*

Below are additional comments/suggestions. Note: line numbers in comments are based on the text version with track changes.

Line 41 "3 to 5 times" for what scenario? If Na were consistent between night and day? Or if sigmaw were consistent? Something else?

*Response: The increase refers to the increase of expected droplet number compared to what it would be based on Na changes alone. This is now clarified in the text.*

Line 120 could you provide some evidence or cite a relevant publication with evidence indicating that assuming the aerosol are internally mixed is a reasonable assumption.

*Response: Relevant references are now added.*

Line 160 – I understand droplets are only formed for positive updraft velocities, but is sigma w calculated from the entire vertical velocity distribution (or just positive velocities)? You mention the gaussian distribution has a mean of zero (which mean there are negative and positive values), but you also say you only use the positive updraft velocities, so it's not clear.

*Response: Only the positive half of the Gaussian distribution around a mean of zero were used for the sigma w calculations. This is now clarified in the text.*

Line 178-180 – reference table 3          *Response: Done.*

Line 245 "These calculations help understand" - suggest rewording          *Response: Amended.*

Line 256 – change 'low' to 'lowest'          *Response: Amended.*

Line 307 – its driven mainly by Na (very little by kappa). I suggest changing 'characteristics' to 'concentrations' and note kappa had little influence (same for the following sentence).

*Response: Amended.*

Line 317-321 this was already mentioned earlier in this section. I suggest removing these last 3 sentences.

*Response: Indeed, these sentences are now removed.*

Line 352 – suggest changing 'strong' to 'high'    *Response: Done.*

Line 358- change 'does' to 'do' and change 'important' to 'large'    *Response: Done.*

Line 419 – add 'of' before 'a'    *Response: Done.*

Table 2. reference equation 2 as the equations used to determine the contributions.

*Response: Done.*

Figure 1. What measurement or model was used to determine BL height.

*Response: The boundary layer height for daytime in the area (1200 m) was taken from Wagner et al. (2015) as calculated during the same measurement campaign while for nighttime the value of 500 m was used from Seidel et al. (2012). This is now clarified in the text.*

Below please find the revised manuscript with marked track changes

[revised manuscript text omitted]